# Efficacy of β-lactam/β-lactamase inhibitor combination is linked to WhiB4-mediated changes in redox physiology of *Mycobacterium tuberculosis*

Saurabh Mishra[1], Prashant Shukla[1,2], Ashima Bhaskar[3], Kushi Anand[1], Priyanka Baloni[4,5], Rajiv Kumar Jha[1], Abhilash Mohan[4], Raju S Rajmani[1], Valakunja Nagaraja[1,6], Nagasuma Chandra[4], Amit Singh[1]*

[1]Microbiology and Cell Biology, Centre for Infectious Disease Research, Indian Institute of Science, Bangalore, India; [2]International Centre for Genetic Engineering and Biotechnology, New Delhi, India; [3]National Institute of Immunology, New Delhi, India; [4]Department of Biochemistry, Indian Institute of Science, Bangalore, India; [5]Molecular Biophysics Unit, Indian Institute of Science, Bangalore, India; [6]Jawaharlal Nehru Centre for Advanced Scientific Research, Bangalore, India

**Abstract** *Mycobacterium tuberculosis* (*Mtb*) expresses a broad-spectrum β-lactamase (BlaC) that mediates resistance to one of the highly effective antibacterials, β-lactams. Nonetheless, β-lactams showed mycobactericidal activity in combination with β-lactamase inhibitor, clavulanate (Clav). However, the mechanistic aspects of how *Mtb* responds to β-lactams such as Amoxicillin in combination with Clav (referred as Augmentin [AG]) are not clear. Here, we identified cytoplasmic redox potential and intracellular redox sensor, WhiB4, as key determinants of mycobacterial resistance against AG. Using computer-based, biochemical, redox-biosensor, and genetic strategies, we uncovered a functional linkage between specific determinants of β-lactam resistance (e.g. β-lactamase) and redox potential in *Mtb*. We also describe the role of WhiB4 in coordinating the activity of β-lactamase in a redox-dependent manner to tolerate AG. Disruption of WhiB4 enhances AG tolerance, whereas overexpression potentiates AG activity against drug-resistant *Mtb*. Our findings suggest that AG can be exploited to diminish drug-resistance in *Mtb* through redox-based interventions.

*For correspondence: asingh@mcbl.iisc.ernet.in

**Competing interests:** The authors declare that no competing interests exist.

## Introduction

*Mycobacterium tuberculosis* (*Mtb*) displays tolerance to several clinically important antibacterials such as aminoglycosides and β-lactams (*Flores et al., 2005a*; *Morris et al., 2005*). Innate resistance of *Mtb* toward β-lactams is likely to be due to the presence of a broad-spectrum Ambler class A β-lactamase (BlaC) (*Flores et al., 2005b*). Other physiological mechanisms such as cell envelope permeability, induction of drug efflux pumps, and variations in peptidoglycan (PG) biosynthetic enzymes may also play a role in the β-lactam-resistance of *Mtb* (*Gupta et al., 2010*; *Lun et al., 2014*). The Ambler class A β-lactamases are mostly susceptible to inhibition by clavulanate (Clav), sulbactam (Sub), and tazobactam (Taz) (*Kurz et al., 2013*). Indeed, intrinsic resistance of *Mtb* toward β-lactams can be overcome by combining β-lactams with Clav (*Chambers et al., 1998*; *Hugonnet et al., 2009*). The combined amoxicillin (Amox) and Clav preparation, referred to as Augmentin (AG), was not only active against *Mtb* in vitro, but also had significant early bactericidal activity in patients with drug-resistant TB (*Chambers et al., 1998*; *Cynamon and Palmer, 1983*). Furthermore, a

**eLife digest** A bacterium called *Mycobacterium tuberculosis* causes tuberculosis in humans. Multiple antibiotics are available to treat this infection, yet around one million people still die from tuberculosis each year. One of the reasons that the number of deaths is so high is because many *M. tuberculosis* cells have become resistant to these drugs. Therefore, new drug treatments are urgently needed to tackle the disease.

When cells are under stress – for example, when a bacterial cell is exposed to an antibiotic – they can increase the production of chemicals known as reactive oxygen species. These chemicals are vital to many processes in cells, but if their levels get too high they can kill cells by damaging DNA and other molecules. To prevent this damage, bacterial cells produce molecules, such as mycothiol, to neutralize the excess reactive oxygen species.

A therapy called Augmentin is used to fight many different types of bacterial infection. It combines an antibiotic known as amoxicillin with another drug that blocks the activity of a bacterial enzyme responsible for breaking down amoxicillin-like drugs. Augmentin can also kill *M. tuberculosis* cells, but it was not clear exactly how it works, or how the bacteria might be able to develop resistance to this treatment. Here, Mishra et al. combined a computational technique known as network analysis with experiments to study the affect of Augmentin on *M. tuberculosis*.

The experiments reveal that *M. tuberculosis* cells can develop resistance to Augmentin by increasing the production of an enzyme that breaks down the antibiotic and by neutralizing reactive oxygen species with help of mycothiol. Augmentin treatment can decrease the production of a protein called WhiB4 in the bacteria. This protein is involved in detecting when cells are stressed and regulates the levels of both mycothiol and the enzyme that breaks down the antibiotic. Increasing the production of this protein made the bacterial cells more susceptible to Augmentin treatment by decreasing the levels of active mycothiol and reducing the production of the enzyme that breaks down the antibiotic drug.

These findings suggest that Augmentin could be more effective against drug-resistant tuberculosis and other bacterial infections if it is combined with a drug that can alter the levels of reactive oxygen species inside bacterial cells. The next step is to search for new molecules that may be able to perform such a role.

combination of meropenem and Clav showed significant bactericidal activity against drug-resistant strains of *Mtb* (*Hugonnet et al., 2009*). In view of this, there is an imminent need to investigate the mechanisms of action of $\beta$-lactams in combination with Clav against *Mtb*, and the potential development of resistance by the pathogen against this combination.

In other bacteria, $\beta$-lactams directly interact with enzymes involved in PG synthesis. This is likely to result in killing of the pathogen through multiple mechanisms, including the induction of autolysin pathway, holin:antiholin pathway, DNA damage, and alterations in physiology (e.g. TCA cycle and oxidative stress) (*Tomasz, 1974*; *Rice et al., 2003*; *Miller et al., 2004*; *Kohanski et al., 2007*; *Lobritz et al., 2015*). The complex effects of $\beta$-lactams on both PG biosynthesis and other processes indicate that the response to $\beta$-lactams could be mediated either through direct sensing of $\beta$-lactam molecules or by their effects on bacterial physiology. In *Staphylococcus aureus,* a transmembrane protease (BlaR1) senses $\beta$-lactam concentrations by direct binding through an extracellular domain, which activates its intra-cytoplasmic proteolytic domain resulting in cleavage of the $\beta$-lactamase repressor, BlaI, and induction of $\beta$-lactamase expression (*Gregory et al., 1997*). It has been shown that *Mtb* expresses a homolog of BlaR1 (encoded by Rv1845c, *blaR*), which modulates the activity of BlaC by regulating the BlaI repressor in a manner analogous to *S. aureus* BlaR1-BlaI couple (*Sala et al., 2009*). However, BlaR orthologues in all mycobacterial species lack the extracellular sensor domain involved in binding with $\beta$-lactams (*Sala et al., 2009*), indicating that mechanisms of antibiotic sensing and BlaC regulation are likely to be distinct in *Mtb*. Furthermore, how $\beta$-lactams influence mycobacterial physiology (e.g. redox balance and primary metabolism) remains unknown. Therefore, insights on how the presence of $\beta$-lactams is conveyed in *Mtb* to activate appropriate adaptation response are key to combating resistance and developing novel therapies.

In this work, we generated a system-scale understanding of how AG affects mycobacterial physiology. Exploiting a range of technologies, we explained mechanistically that the efficacy of AG is partly dependent upon the redox physiology of *Mtb*. Furthermore, we have rationally described the role of a redox-responsive transcription factor, WhiB4, in regulating the tolerance of *Mtb* to AG during infection. Our study demonstrates how *Mtb* alters its redox physiology in response to AG and identifies a major mycobacterial antioxidant, mycothiol (MSH), and WhiB4 as major contributors to β-lactam tolerance.

## Results

### Network analysis revealed modulation of cell wall processes in response to AG in *Mtb*

To assess the response of *Mtb* toward β-lactam and β-lactamase inhibitor combination(s), we analyzed the transcriptome of mycobacterial cells exposed to AG. We observed that 100 μg/ml of Amox in combination with 8 μg/ml of Clav (10X MIC of AG) arrested bacterial growth at 6 hr and killing was observed only after 12 hr post-exposure (*Figure 1—figure supplement 1*-Inset). Therefore, expression changes at a pre-lethal phase (i.e. 6-hr post -exposure) can reveal significant insights into *Mtb* pathways involved in AG tolerance.

A total of 481 genes were induced (≥2 fold; p value ≤ 0.05) and 461 were repressed (≥2 fold; p value ≤ 0.05) in wt *Mtb* upon AG-treatment (*supplementary file 1A*). Although these results are important, the transcriptome only provides a snapshot of the mechanisms exploited by *Mtb* for AG tolerance. To generate a system-scale understanding, computational approaches that combine condition-specific expression data with general protein interaction data are frequently utilized to construct dynamic and stress response networks (see Appendix 1 for detailed explanation). Therefore, we further generated the AG response network by combining microarray data with the protein-protein interaction (PPI) map of *Mtb*. To construct this map, we first created a comprehensive PPI of *Mtb* using information from experimentally validated and published interactions (see Materials and methods) and integrated microarray data with the PPI to generate the AG response network. In the network, a node represents a protein whose weight is based on a weighting function that captures the variation in the expression level of the corresponding gene due to drug exposure. An edge represents an interaction between two nodes, which are also weighted by a function that captures the node weights of both nodes forming an edge, as a relative importance of all edges in the network. A full description of the mathematical equations and algorithms used to generate the AG response network is beyond the scope of this study, we encourage readers to refer our original papers for detailed methodology (*Sambarey et al., 2013*; *Sambaturu et al., 2016*; *Padiadpu et al., 2016*).

*Figure 1—figure supplement 1* and *supplementary file 1B* represent the top 1% nodes, which cover a total of 806 genes, connected through 1096 interactions to form a well-connected AG response network of *Mtb*. Genes belonging to diverse functional classes such as intermediary metabolism, cell wall, lipid metabolism, virulence, and information pathways are featured in the response network. In line with cell surface targeting activity of β–lactams, the cumulative node weight (CNW) of genes belonging to cell-wall-related processes, including PG biosynthesis, was the highest (CNW = 30616361.02) amongst the classes affected by AG (*Figure 1A*). Interestingly, nodes belonging to 'intermediary metabolism and respiration' were also significantly enriched in response to AG (CNW = 20716788.92; *Figure 1A*), indicating a downstream effect of target-specific interactions of AG on fundamental metabolic processes in *Mtb*. Further analysis revealed that several mediators (e.g. *sigE*, *sigB*, *mprAB*, and *dnaK*) of cell envelope stress response (*Bretl et al., 2014*) function as major hub nodes and form-interconnected networks of genes important for maintaining cell wall integrity in response to AG (*Figure 1—figure supplement 2*). Accordingly, expression data showed induction of genes involved in PG biosynthesis, β-lactamase regulation (*blaR-blaI*), and cell envelope homeostasis in response to AG (*Figure 1B*). In addition, two other mechanisms involved in tolerance toward β-lactams that is outer membrane permeability (mycolic acid biogenesis [*kasA*, *kasB*, and *fabD*] and *omp*) and drug efflux pumps (*efpA*, *Rv1819c* and *uppP*) were also induced (*Figure 1B* and *supplementary file 1A*). Altogether, *Mtb* responds to AG by modulating the expression of cell-envelope-associated pathways including those that are the specific targets of β-lactams.

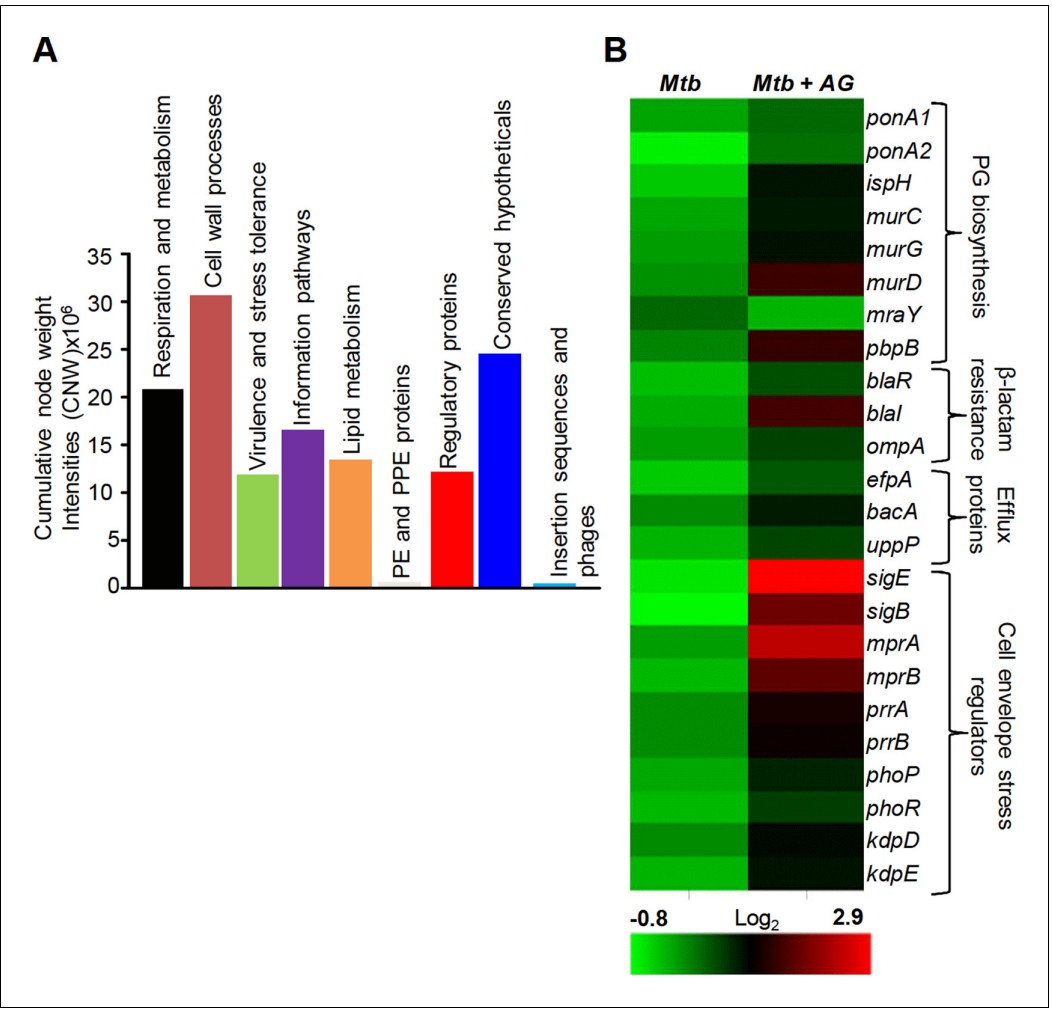

**Figure 1.** Network analysis identified pathways affected by AG exposure in *Mtb*. Wt *Mtb* was grown to an $OD_{600}$ of 0.4 and treated with 100 µg/ml of Amox and 8 µg/ml of Clav (10X MIC of AG) for 6 hr at 37°C. Total RNA was isolated and processed for microarray analysis as described in Materials and methods. (A) Cumulative node weight intensities (CNW) were derived by addition of the node weights of genes in a particular functional group upon exposure to AG. Node weight intensity of a gene was derived by multiplying the normalized intensity value with the corresponding fold-change (FC) value. Cumulative node weight intensities for different functional classes are available in *Figure 1—source data 1*. (B) Heat map showing expression of genes ($\log_2$fold-change, $p \leq 0.05$) that belong to cell wall processes for untreated and AG-treated *Mtb* from two biological samples.

The following source data and figure supplements are available for figure 1:

**Source data 1.** Cumulative node weight intensities for different functional classes as depicted in *Figure 1A*.

**Figure supplement 1.** Global network of *Mtb* under AG stress.

**Figure supplement 2.** Sub-network of major hub nodes showing the top-most activities regulating response of *Mtb* upon AG treatment.

## AG affects pathways associated with central carbon metabolism (CCM), respiration, and redox balance

Since nodes coordinating 'intermediary metabolism and respiration' were the second most enriched class in response to AG, we performed a detailed examination of genes altered in this category. We found that energetically efficient respiratory complexes such as NADH dehydrogenase I (*nuo*

operon) and ATP-synthase (*atpC, atpG,* and *atpH*) were down-regulated, whereas energetically less favored NADH dehydrogenase type II (*ndh*), cytochrome *bd* oxidase (*cydAB*), and nitrite reductase (*nirBD*) were activated in response to AG (*Figure 2*). The transcriptional shift toward a lesser energy state is consistent with the down-regulation of several genes associated with the TCA cycle (*sucCD, fum, mdh,* and *citA*), along with an induction of glycolytic (*pfkA, pfkb, fba,* and *pgi*), gluconeogenesis (*pckA*), and glyoxylate (*icl1*) pathways (*Figure 2*). Interestingly, *icl1* has recently been shown to promote tolerance of *Mtb* toward diverse anti-TB drugs by maintaining redox homeostasis (*Nandakumar et al., 2014*). These findings indicate that the maintenance of redox balance is likely to be an important cellular strategy against AG.

The fact that there was a significant upregulation of genes involved in oxidative stress response in *Mtb* was indicative of the influence of AG on mycobacterial redox physiology. We found increased expression of reactive oxygen species (ROS) detoxifying enzymes (*ahpCD, katG,* and *hpx*), antioxidant buffers (*trxB1, trxA, trxC,* and *mtr*), methionine sulfoxide reductase (*msrA*), Fe-S cluster repair system (*Rv1461-Rv1466; suf* operon), and intracellular redox sensors (*whiB6, whiB2, whiB3,* and *pknG*) (*Figure 2*). The global regulator of oxidative stress in bacteria (OxyR) is non-functional in *Mtb* (*Deretic et al., 1995*). However, we had earlier reported that a redox-sensitive DNA-binding protein (WhiB4) functions as a negative regulator of OxyR-specific antioxidant genes (e.g. *ahpCD*) in *Mtb* (*Chawla et al., 2012*). Consequently, *Mtb* lacking *whiB4* (*MtbΔwhiB4*) displayed higher expression of antioxidants and greater resistance toward oxidative stress (*Chawla et al., 2012*). While a modest repression of WhiB4 (~1.3 fold) in response to AG was observed in microarray experiments, qRT-PCR analysis showed a significant down-regulation ($-5.00 \pm 0.27$ fold; p value $\leq 0.001$) as compared to unstressed *Mtb* (*Figure 2—figure supplement 1*). The breakdown of iron homeostasis is another hallmark of oxidative stress (*Imlay, 2003*). Accordingly, our data exhibited induction of two Fe-responsive repressors (*ideR* and *furB*) along with the down-regulation of genes encoding Fe-siderophore biosynthetic enzymes (*mbt* operon) and Fe-transport (*Rv1348*), and up-regulation of Fe-storage (*bfrB*) (*Figure 2*).

Recently, two mycobacterial redox buffers, mycothiol (MSH) and ergothionene (EGT), were implicated in protection against oxidants and antibiotics (*Saini et al., 2016*). We compared gene expression changes displayed by MSH and EGT mutants (*Saini et al., 2016*) with the AG transcriptome. Approximately 60% of genes regulated by MSH and EGT also displayed altered expression in response to AG (*Figure 2—figure supplement 2*), indicating overlapping roles of MSH and EGT in tolerating oxidative stress associated with AG treatment in *Mtb* (*Saini et al., 2016*). Lastly, we performed transcriptomics of *Mtb* in response to a known oxidant cumene hydroperoxide (CHP; 250 μM for 2 hr [non-toxic concentration]) and compared with expression changes induced by AG. As shown in *Figure 2—figure supplement 3, a* considerable overlap in gene expression (~30%) was observed between these two conditions (*supplementary file 1C*). More importantly, genes associated with β-lactam tolerance (*ponA2, ispH, blaR,* and *kasA*) and redox-metabolism (*ahpCD, trxB1, trxB2, trxC,* and *suf*) were similarly regulated under CHP and AG challenge (*supplementary file 1C*).

It can be argued that the high concentration of AG (10X MIC) can adversely affect *Mtb* physiology to influence primary response of AG on gene expression. To address this issue, we reassessed global changes in gene expression upon exposure to 1X and 5X MIC of AG at 6 hr and 12 hr post-treatment. A relatively small number of genes were differentially regulated by lower concentrations of AG as compared to 10X MIC (*supplementary file 1D*). However, similar to our results using 10X MIC, we found that exposure to 1X and 5X MIC of AG increased expression of genes associated with PG biogenesis (*mur operon, ponA1,* and *ponA2*), β-lactamase regulation (*blaI-blaR*), cell envelope stress (*sigB, sigE, mprAB,* and *phoPR*), redox metabolism (*hpx, trx system, msrA, suf operon, whiBs,* and *pknG*), alternate respiration (*ndh* and *cydAB*), CCM (*pfkAB, fba, icl,* and *aceAa*), and efflux pumps (*efpA* and *uppP*) (*Figure 2—figure supplement 4*). Lastly, we validated our microarray data by performing qRT-PCR on a few genes highly deregulated upon treatment with 1X, 5X, and 10X MIC of AG (*Figure 2—figure supplement 1*). Taken together, these data indicate a major recalibration of genes regulating cell wall processes and cellular bioenergetics of *Mtb* in response to AG.

## AG treatment induces redox imbalance in *Mtb*

Altered expression of genes associated with respiration and oxidative stress response indicates that AG exposure might elicit redox stress in *Mtb*. To investigate this, we performed a comprehensive evaluation of changes in redox physiology of *Mtb* upon exposure to AG. Since, NADH redox

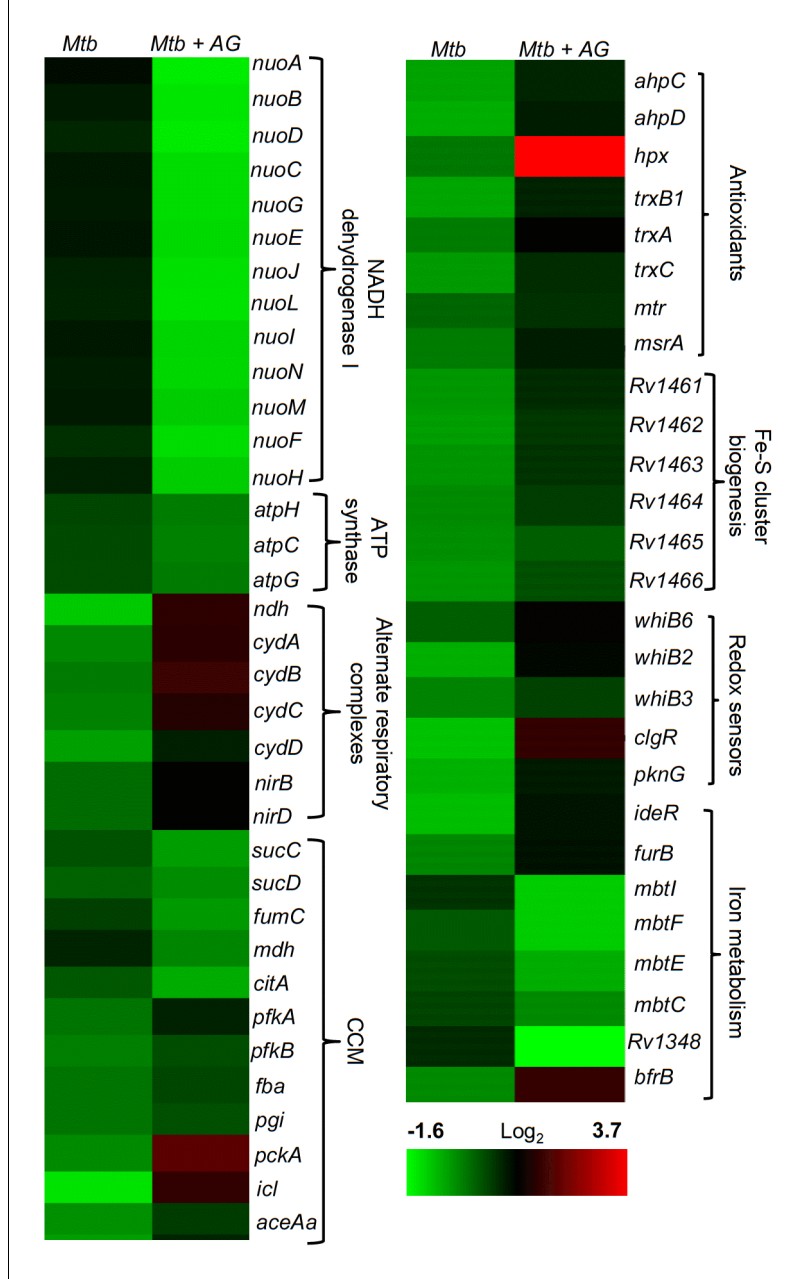

**Figure 2.** AG influences multiple pathways involved in central metabolism, respiration and redox balance in *Mtb*. Heat maps depicting expression of genes (log$_2$fold-change; p≤0.05) coordinating respiration, CCM, iron-metabolism and redox balance for untreated and 6 hr of AG-treated *Mtb* from two biological samples.

The following figure supplements are available for figure 2:

**Figure supplement 1.** qRT-PCR analysis of *Mtb* exposed to different concentrations of AG for indicated time points.

**Figure supplement 2.** Comparative analysis of genes differentially regulated by AG treatment and upon depletion of mycothiol or ergothioneine buffers.

**Figure supplement 3.** Overlapping regulation of genes in response to AG and oxidative stress.

*Figure 2 continued on next page*

*Figure 2 continued*

**Figure supplement 4.** Heat maps depicting gene expression profile (log$_2$fold-change) of *Mtb* untreated or treated with 1X and 5X MIC of AG for 6 and 12 hr from at least three biological samples.

cofactor is central to metabolism and respiration, we first measured NADH/NAD$^+$ ratio of *Mtb* cells exposed to 10X MIC of AG at various time points post-treatment. At pre-lethal stage (6 hr post-treatment), we did not observe any change in NADH/NAD$^+$ ratios (*Figure 3A*). However, a significant elevation of NADH/NAD$^+$ ratio was detected 24 hr post-treatment, which coincides with AG-induced killing in *Mtb* (*Figure 3A*). We subsequently determined accumulation of ROS by staining with an oxidant-sensitive fluorescent dye; 2′,7′-dichlorofluorescein diacetate (DCFDA) in *Mtb* cells treated with AG (10X MIC) for 3 hr and 6 hr. Early time points were considered for ROS measurements to disregard the possibility of death-mediated increase in ROS upon AG treatment. A consistent increase (~3-fold increase) in DCFDA fluorescence was observed at both time points as compared to untreated control (*Figure 3B*). Under aerobic conditions, ROS is mainly generated through univalent reduction of O$_2$ by reduced metals, flavins, and quinones (*Imlay, 2013*), which mainly generates superoxide (O$_2^{-\bullet}$). Therefore, we determined O$_2^{-\bullet}$ production using a well-established and freely cell-permeable O$_2^{-\bullet}$ indicator, dihydroethidium (DHE) (*Kalyanaraman et al., 2014*). It is known that DHE specifically reacts with O$_2^{-\bullet}$ to release fluorescent product 2-hydroxyethidium (2-OH-E$^+$), which can be conveniently detected by HPLC (*Kalyanaraman et al., 2014*). The reaction of DHE with other oxidants produces ethidium (E$^+$) (*Tyagi et al., 2015*). Due to biosafety challenges associated with a BSL3 category pathogen such as *Mtb* for HPLC, we measured O$_2^{-\bullet}$ levels inside the related but nonpathogenic *Mycobacterium bovis* BCG upon AG challenge. BCG cells were treated with AG (10X MIC) for 3 hr and 6 hr, followed by DHE staining and HPLC. We found that BCG cells treated with AG generate peaks corresponding to O$_2^{-\bullet}$ (2-OH-E$^+$) and other ROS (E$^+$) (*Figure 3C*). The intensity of peaks was significantly higher at 6 hr post-treatment as compared to untreated control (*Figure 3C*). As a control, we used a well-known O$_2^{-\bullet}$ generator (menadione) in our assay and similarly detected a 2-OH-E$^+$ peak (*Figure 3C*, inset). Thereafter, we determined whether the thiol-based antioxidant thiourea can reverse the influence of AG on viability of *Mtb*. Thiourea has recently been shown to protect *Mtb* from oxidative stress by modulating the expression of antioxidant genes (*Nandakumar et al., 2014*). *Mtb* was co-incubated with various concentrations of thiourea and AG, and viability was measured after 10 days. Thiourea did not exert a

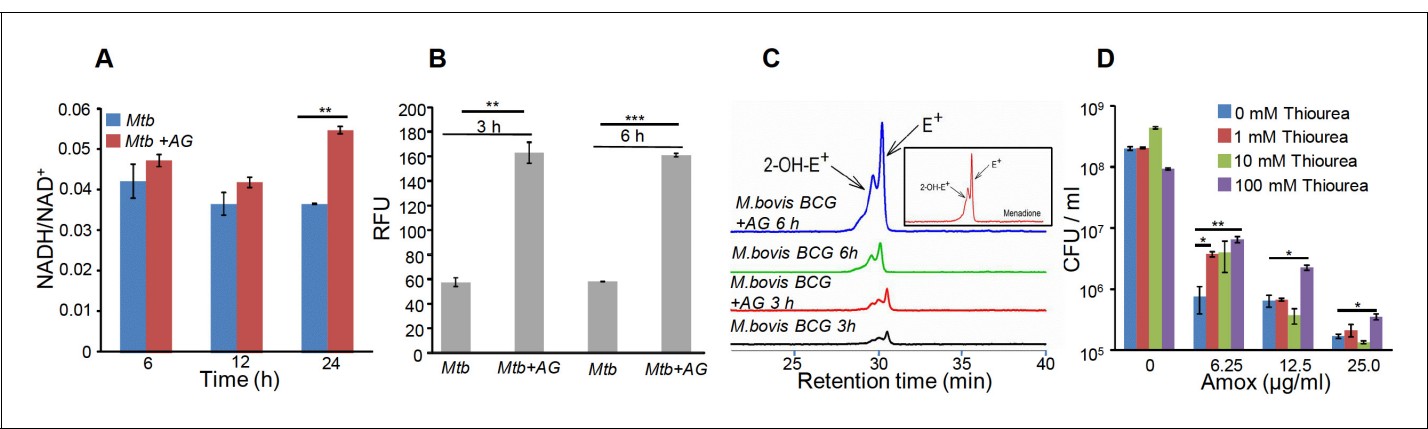

**Figure 3.** AG influences the internal redox physiology of *Mtb*. Wt *Mtb* or *M. bovis* BCG was grown to OD$_{600}$ of 0.4 and treated with 10X MIC of AG). At indicated time points, cells were analyzed for (A) NADH/NAD$^+$ estimation, (B) ROS measurement using oxidant-sensitive fluorescent dye; 2′,7′-dichlorofluorescein diacetate (DCFDA), and (C) Superoxide estimation using dihydroethidium (DHE) as described in Materials and methods. (D) Wt *Mtb* was grown as described earlier and exposed to specified concentrations of Amox in the presence of 8 µg/ml of Clav for 10 days in the presence or absence of thiourea and survival was measured using colony-forming unit (CFU) counts. Error bars represent standard deviations from the mean. *p≤0.05, *p≤0.01 and ***p≤0.001. Data are representative of at least two independent experiments done in duplicate.

significant effect on the survival of *Mtb* under normal growing conditions (*Figure 3D*); however, it did increase the survival of *Mtb* treated with 0.625X and 1.25X MIC of AG by ~10- and 5-folds, respectively (*Figure 3D*). At higher AG concentrations (2.5X MIC), only 100 mM of thiourea showed a twofold protective effect (*Figure 3D*).

The above data indicate that bactericidal consequences of AG may be dependent upon internal oxidant-antioxidant balance of *Mtb*. To demonstrate this unambiguously, we exploited a mycobacterial-specific non-invasive biosensor (Mrx1-roGFP2) to measure the redox potential of a physiologically relevant and abundant cytoplasmic antioxidant, MSH (*Bhaskar et al., 2014*). Any changes in the oxidation-reduction state of MSH can be reliably quantified by ratiometric measurement of emission at 510 nm after excitation at 405 and 488 nm (*Bhaskar et al., 2014*). *Mtb* expressing Mrx1-roGFP2 was treated with lower (0.2X MIC) and higher (10X MIC) concentrations of AG and intramycobacterial $E_{MSH}$ was determined by measuring biosensor ratiometric response over time, as described previously (*Bhaskar et al., 2014*). We observed a modest but consistent increase in 405/488 ratio at 6 hr and 24 hr post-treatment with 10X and 0.2X MIC of AG, respectively (*Figure 4A*), indicating that antioxidant mechanisms are mostly efficient in minimizing the impact of AG-mediated ROS generation on internal $E_{MSH}$ of *Mtb*.

Importantly, to determine whether AG-induced oxidative stress is physiologically relevant in the context of infection, we measured dynamic changes in $E_{MSH}$ of *Mtb* inside human macrophage cell line (THP-1) during infection. Infected macrophages were exposed to AG (1.25-fold to 10-fold of the in vitro MIC) and the redox response was measured by flow cytometry. As reported earlier, *Mtb* cells inside macrophages displayed variable $E_{MSH}$, which can be resolved into $E_{MSH}$-basal (−270 mV), $E_{MSH}$-oxidized (−240 mV), and $E_{MSH}$-reduced (−310 mV) subpopulations (*Bhaskar et al., 2014*). Treatment with AG induces significant increase in the oxidized subpopulation over time (*Figure 4B*). In parallel, we examined whether the elevated oxidative stress correlates with the killing potential of AG during infection. Macrophages infected with *Mtb* were treated with 10X MIC of AG and bacillary load was monitored by enumerating colony-forming units (CFUs) at various time points post-infection. At 6 hr and 12 hr post-AG treatment, the effect on *Mtb* survival was marginal (*Figure 4C*). However, an ~100-fold decline in CFU was observed at 24 hr and 36 hr post-AG treatment (*Figure 4C*). More importantly, an increase in $E_{MSH}$-oxidized subpopulation was observed at a time point where survival was not considerably affected (6 hr) (*Figure 4B and C*). This suggests that AG-mediated oxidative stress precedes bacterial death inside macrophages and that the intramycobacterial oxidative

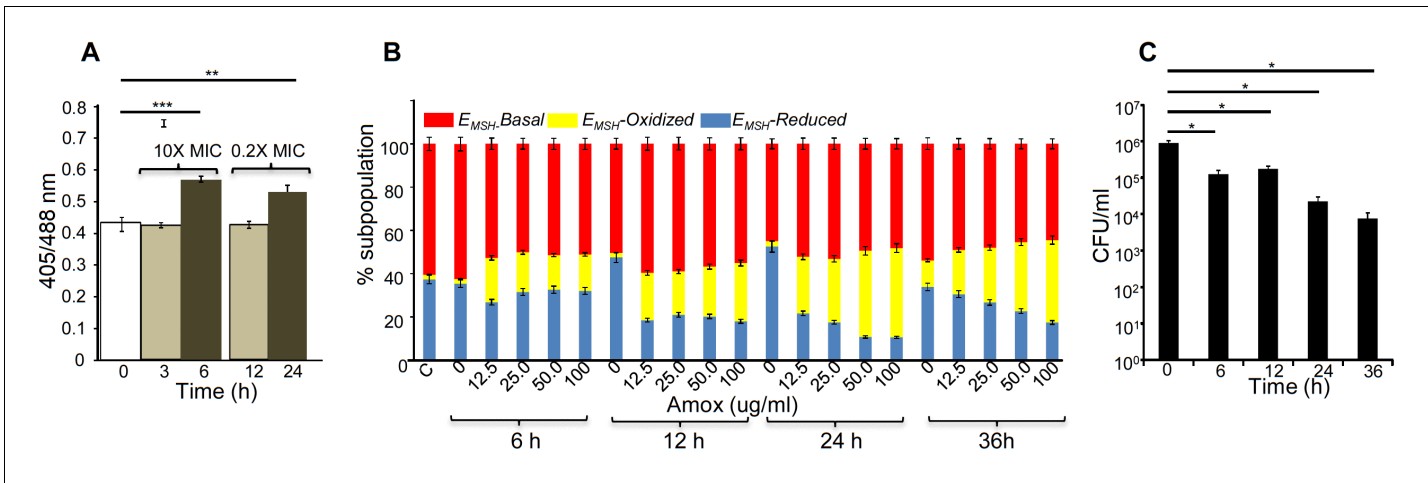

**Figure 4.** AG induces oxidative shift in $E_{MSH}$ of *Mtb* in vitro and during infection. (**A**) Wt *Mtb*-expressing Mrx1-roGFP2 was treated with lethal (10X MIC) and sub-lethal (0.2 X MIC) concentrations of AG and ratiometric sensor response was measured at indicated time points by flow cytometry. (**B**) PMA-differentiated THP-1 cells were infected with *Mtb* expressing Mrx1-roGFP2 (moi: 10) and treated with indicated concentrations of Amox in the presence of 8 μg/ml of Clav as described in Materials and methods. At the indicated time points, ~30,000 infected macrophages were analyzed by flow cytometry to quantify changes in *Mtb* subpopulations displaying variable $E_{MSH}$ as described in Materials and methods. (**C**) In parallel experiments, infected macrophages were lysed and bacillary load was measured by plating for CFU. Error bars represent standard deviations from the mean. *p≤0.05, **p≤0.01 and ***p≤0.001. Data are representative of at least two independent experiments done in duplicate.

stress is not a consequence of AG-induced toxicity. Altogether, our data showed that AG perturbs mycobacterial redox physiology and the environment inside macrophages potentiates the mycobactericidal effect of AG.

## Mycothiol buffer protects *Mtb* from AG-mediated killing

Since AG induces intramycobacterial oxidative stress, it is likely that the loss of major intracellular antioxidant, MSH, might potentiate the antimycobacterial activity of AG. To examine this, we used a MSH-negative strain (*MsmΔmshA*) (*Rawat et al., 2002, 2007*) of *Mycobacterium smegmatis (Msm)*, an organism that is widely used as a surrogate for pathogenic strains of *Mtb*. Wt *Msm* and *MsmΔmshA* strains were exposed to various concentrations of Amox at a saturating concentration of Clav (8 μg/ml) and percent growth inhibition was measured using the Alamar blue (AB) assay. AB is an oxidation-reduction indicator dye which changes its color from non-fluorescent blue to fluorescent pink upon reduction by actively metabolizing cells, whereas inhibition of growth by antimycobacterial compounds interferes with AB reduction and color development (*Tyagi et al., 2015*). As shown in *Figure 5A*, at a fixed Clav concentration, *MsmΔmshA* exhibited ~3 and 10-fold higher inhibition at 5 μg/ml and 2.5 μg/ml of Amox as compared to wt *Msm*, respectively. At 10 μg/ml of Amox, both strains showed nearly complete inhibition (*Figure 5A*). Next, we measured susceptibility to Clav at a fixed concentration of Amox (10 μg/ml). Higher concentrations of Clav (10 μg/ml) inhibited the growth of *Msm* and *MsmΔmshA* with a comparable efficiency (*Figure 5B*). However, while wt *Msm* overcomes the inhibitory effect of Amox at lower Clav concentrations, *MsmΔmshA* remained sensitive to Amox even at the lowest concentration of Clav (0.625 μg/ml) (*Figure 5B*). As shown in *Figure 5B*, *MsmΔmshA* exhibited ~7-fold greater inhibition at 0.625 μg/ml of Clav as compared to *Msm*. We further validated the contribution of MSH in tolerating AG by measuring the sensitivity of *Msm* lacking MSH-disulphide reductase (Mtr) activity (*MsmΔmtr*) (*Holsclaw et al., 2011*) and MSH-depleted (*MsmΔmshD*) strain toward Amox and AG. A twofold reduction in MIC for Amox and AG was detected in case of *MsmΔmtr* as compared to wt *Msm*, whereas *MsmΔmshD* remained unaffected (*Table 1*). Since *MsmΔmshD contains* only ~3% of total cellular MSH but accumulates two novel thiols (Suc-MSH and formyl-MSH) (*Newton et al., 2005*), our data suggest that Msm can also alleviate redox stress caused by AG via Suc-MSH and/or formyl-MSH. Alongside MSH, other prominent oxidative stress defense mechanisms include the $H_2O_2$ detoxifying enzyme, catalase (KatG), and NADPH-dependent thioredoxin (TRX) system. The extra-cytoplasmic sigma factor, SigH, is known to regulate several components of the TRX system in mycobacteria (*Raman et al., 2001*).

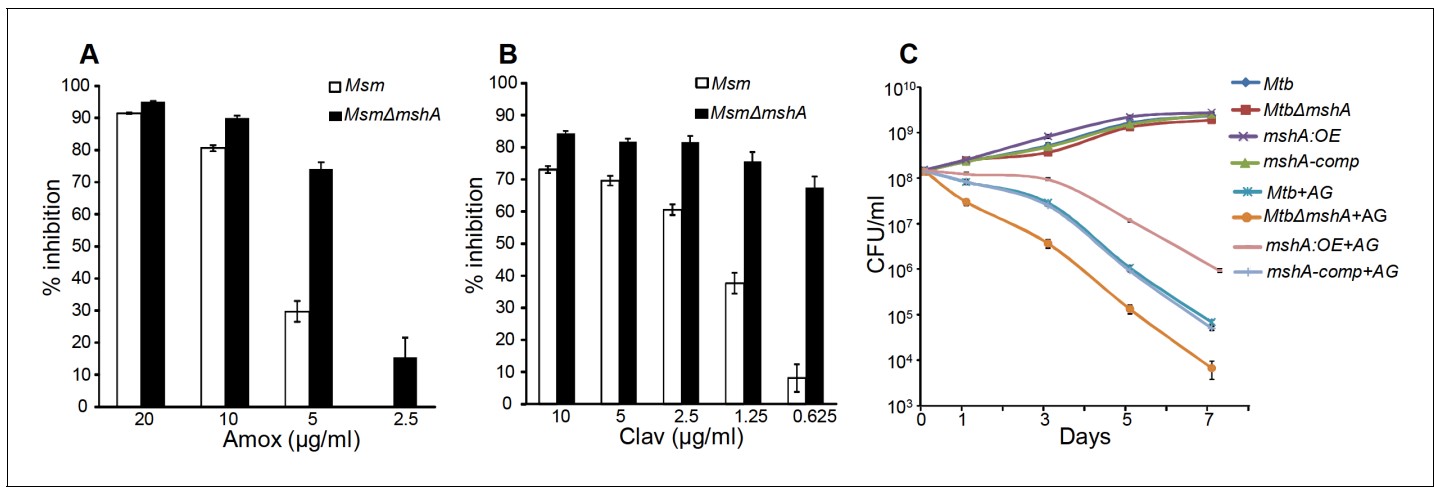

**Figure 5.** Mycothiol mediates tolerance to AG. Wt *Msm* and *MsmΔmshA* strains were grown to OD$_{600}$ of 0.4 and either treated with various concentrations of (**A**) Amox and Clav (8 μg/ml) or (**b**) Clav and Amox (10 μg/ml) and % inhibition in growth was measured by Alamar blue (AB) assay as described in Materials and methods. (**C**) Wt *Mtb*, *MtbΔmshA*, *mshA-comp* and *mshA:OE* strains were exposed to 10X MIC of AG and survival was monitored by measuring CFU over time. Error bars represent standard deviations from mean. Data are representative of at least two independent experiments done in duplicate.

**Table 1.** Minimum inhibitory concentrations (MICs) of Amox and AG for different *Mycobacterium smegmatis* strains. Source data file containing the images for MIC calculation is available in **Table 1–source data 1**.

| | μg/mL | |
|---|---|---|
| **Strains** | **Amox** | **Amox+clav (AG)** |
| wt *Msm* | 160 | 20 + 8 |
| *MsmΔmtr* | 80 | 10 + 8 |
| *MsmΔmshD* | 160 | 20 + 8 |
| *MsmΔkatG* | 80 | 10 + 8 |
| *MsmΔsigH* | 20 | 10 + 8 |

Source data 1. Images of Alamar blue assay plates for calculation of minimum inhibitory concentration (MIC).

Therefore, we assessed the inhibition of Msm strains lacking KatG (*MsmΔkatG*) (*Padiadpu et al., 2016*) and SigH (*MsmΔsigH*) (*Fernandes et al., 1999*) by Amox and AG. Both strains exhibited a twofold increased susceptibility toward Amox and AG (*Table 1*), further confirming a link between mycobacterial redox physiology and AG efficacy.

To confirm that the above findings can be recapitulated in slow growing pathogenic mycobacteria (i.e. *Mtb* H37Rv), we utilized *mshA*-deficient (*MtbΔmshA*), *mshA*-complemented (*mshA-comp*), and *mshA*-overexpressing (*mshA:OE*) strains of *Mtb*. The *mshA:OE* strain was generated by conditionally overexpressing *mshA* in *Mtb* using an anhydrotetracyline (Atc)-inducible system (TetR) (*Mehta et al., 2016*; *Vilchèze et al., 2008*; *Parikh et al., 2013*). We performed $E_{MSH}$ measurements and confirmed that the overexpression of *mshA* shifted the ambient $E_{MSH}$ of *Mtb* from −275 ± 3 mV to −300 ± 5 mV, indicating an overall elevation in anti-oxidative potential of *Mtb*. The MSH-deficient strain of *Mtb* (*MtbΔmshA*) displayed the oxidative $E_{MSH}$ of >-240 mV, whereas *mshA*-complemented strain (*mshA-comp*) displayed a $E_{MSH}$ comparable to wt *Mtb* (i.e. −275 ± 3 mV). Wt *Mtb*, *mshA:OE*, *MtbΔmshA,* and *mshA-comp* were exposed to 10X MIC of AG and growth was monitored over time by measuring CFUs. AG treatment resulted in a time-dependent decrease in the growth of *Mtb* strains (*Figure 5C*). However, the decline was severe in case of *MtbΔmshA* as compared to wt *Mtb*, whereas *mshA-OE* showed relatively better tolerance than wt *Mtb* (*Figure 5C*). Expression of *mshA* from its native promoter (*mshA-comp*) restored tolerance comparable to wt *Mtb* (*Figure 5C*). In summary, AG exposure triggers the redox imbalance and cellular antioxidants such as MSH provide efficient tolerance toward AG.

### *Mtb* WhiB4 modulates gene expression and maintains $E_{MSH}$ in response to AG

Altered expression of oxidative stress-specific genes, elevation of ROS, and perturbation of $E_{MSH}$ upon AG exposure suggest that intramycobacterial redox potential can serve as an internal cue to monitor the presence of β-lactams. Canonical intracellular redox sensors such as OxyR, SoxR, and FNR are either absent or rendered non-functional in *Mtb* (*Deretic et al., 1995*; *Chawla et al., 2012*). We have previously shown that *Mtb* features a Fe-S cluster-containing transcription factor (WhiB4), which responds to oxidative stress by regulating the expression of antioxidant genes (*Chawla et al., 2012*). Since *whiB4* expression is uniformly repressed by β-lactams (e.g. meropenem and AG) (*Lun et al., 2014*) and oxidative stress (*Chawla et al., 2012*), WhiB4 appears to be critical in the β-lactam-induced oxidative stress response in *Mtb*. We assessed this connection by examining the expression of *whiB4* in MSH-deficient (*MtbΔmshA*) and MSH-sufficient (*mshA-OE* and *mshA-comp*) strains by qRT-PCR. Expression analysis demonstrated that the *whiB4* transcript was significantly repressed in *MtbΔmshA* (−2.94 ± 0.22 fold), whereas expression is restored in *mshA-OE* (−1.08 ± 0.09 fold) and *mshA-comp* (−1.16 ± 0.08), in comparison to wt *Mtb* (*Figure 6—figure*

*supplement 1*). Overall, WhiB4 regulatory function is modulated by the internal redox physiology of *Mtb*.

Based on the above evidence, we tested the direct role of WhiB4 in β-lactam tolerance. We performed microarray analyses of *MtbΔwhiB4* (*Chawla et al., 2017*) upon treatment with AG (at 10X MIC of *Mtb*) for 6 hr as described previously. A total of 495 genes were induced (≥1.5 fold; p value ≤ 0.05) and 423 were repressed (≥1.5 fold, p value ≤ 0.05) in the *MtbΔwhiB4* as compared to wt *Mtb* upon AG-treatment (*supplementary file 2A*). Our network analysis showed that diverse functional classes such as cell wall processes, virulence adaptation pathways, intermediary metabolism and respiration, and lipid metabolism were affected in *MtbΔwhiB4* upon exposure to AG (*Figure 6A*). Microarray data indicated higher expression of genes known to be involved in tolerance to β-lactams in *MtbΔwhiB4*. Transcription of *blaR* and *blaC* was induced 8.43 ± 4.75 and 2.23 ± 0.19 fold, respectively, in *MtbΔwhiB4* as compared to wt *Mtb* upon AG treatment (*Figure 6B*, *Figure 6— figure supplement 2*). Other genetic determinants of β-lactam tolerance such as PG biosynthetic genes (*murE*, *murF*, and *murG*), penicillin-binding proteins (*Rv2864c*, *Rv3627c*, and *Rv1730c*), and cell division and DNA transaction factors (*ftsK* and *fic*) (*Figure 6B*) were also up-regulated in *MtbΔwhiB4* upon treatment. Furthermore, *MtbΔwhiB4* showed greater expression of DNA repair genes (SOS response), many of which are known to interfere with cell division and promote β-lactam tolerance in other bacterial species (*Miller et al., 2004*) (*Figure 6C*). Since transcriptional data implicate WhiB4 in regulating the biosynthesis of the PG polymer, we stained the same from wt *Mtb*, *MtbΔwhiB4*, and *whiB4-OE* cells using a fluorescent derivative of the PG binding antibiotic, vancomycin (Bodipy-VAN) and imaged the cells using confocal microscopy. To generate the *whiB4-OE* strain, we overexpressed WhiB4 using an inducer anhydrotetracycline (Atc), in *MtbΔwhiB4* as described previously (*Chawla et al., 2012*). As expected, poles of wt *Mtb* and *whiB4-OE* cells were fluorescently labeled, consistent with the incorporation of nascent PG at the poles in mycobacteria (*Figure 6—figure supplement 3*) (*Thanky et al., 2007*). Interestingly, Bodipy-VAN was found to label the entire length of *MtbΔwhiB4*, indicating deposition of PG along the entire body of the cylindrical cells (*Figure 6—figure supplement 3A*). *MtbΔwhiB4* cells were also marginally longer than wt *Mtb* (*Figure 6—figure supplement 3B*). Further experimentations are required to understand how WhiB4 modulates PG biosynthesis and cell size. Nonetheless, our transcriptomics and imaging data are in reasonable agreement with each other and support the PG-regulatory function of WhiB4 in *Mtb*.

Other mechanisms that could link WhiB4 with drug tolerance include the heightened expression of cation transporters, ABC-transporters, PDIM lipid biogenesis (*ppsC*, *ppsD*, *ppsE*, *drrA*, *drrB*, and *drrC*), and drug-efflux pumps (*Rv1258c* and *Rv1634*) in *MtbΔwhiB4* upon AG exposure (*Figure 6D*). Lack of the Rv1258c pump and PDIM lipids have been reported to sensitize *Mtb* towards β-lactams and vancomycin (*Dinesh et al., 2013*; *Soetaert et al., 2015*). Several PE_PGRS genes involved in maintaining cell wall architecture and protection from oxidative stresses were up-regulated in *MtbΔwhiB4* (*Figure 6E*) (*Fishbein et al., 2015*). Our results also revealed that redox-metabolism is significantly altered in *MtbΔwhiB4* in response to AG. For example, components of the NADH-dependent peroxidase (*ahpCD*), peroxynitrite reductase complex (*dlaT*), thiol-peroxidase (*tpx*), mycothiol biosynthesis (*mshA*), pyruvate dehydrogenase complex (*pdhA*, *pdhC*, and *aceE*), and hemoglobin like proteins (*glbO*) were induced in AG-challenged *MtbΔwhiB4* compared to wt *Mtb* (*Figure 6F*). Importantly, most of these enzymatic activities are well known to confer protection against oxidative and nitrosative stress in *Mtb* (*Master et al., 2002*; *Hu and Coates, 2009*; *Ung and Av-Gay, 2006*; *Maksymiuk et al., 2015*; *Venugopal et al., 2011*; *Pathania et al., 2002*). Furthermore, similar to wt *Mtb*, primary NADH dehydrogenase complex (*nuo operon*) was down-regulated in *MtbΔwhiB4* in response to AG treatment (*supplementary file 2B*). However, compensatory increase in alternate respiratory complexes such as *ndh* and *cydAB* was notably higher in *MtbΔwhiB4* than in wt *Mtb*, indicating that *MtbΔwhiB4* is better fit to replenish reducing equivalents during drug-induced cellular stress (*Figure 6G*). In tune with this, components of the TCA cycle and pentose phosphate pathway involved in generating cellular reductants (NADH and NADPH) were induced in *MtbΔwhiB4* as compared to wt *Mtb*. We validated our microarray data by performing qRT-PCR on a few genes deregulated upon AG treatment in *MtbΔwhiB4* (*Figure 6—figure supplement 2*).

Overall, AG-exposure elicits transcriptional changes, which are indicative of a higher potential of *MtbΔwhiB4* to maintain redox homeostasis upon drug exposure. We directly assessed this by examining changes in $E_{MSH}$ of *MtbΔwhiB4* and *whiB4-OE* in response to AG in vitro and inside

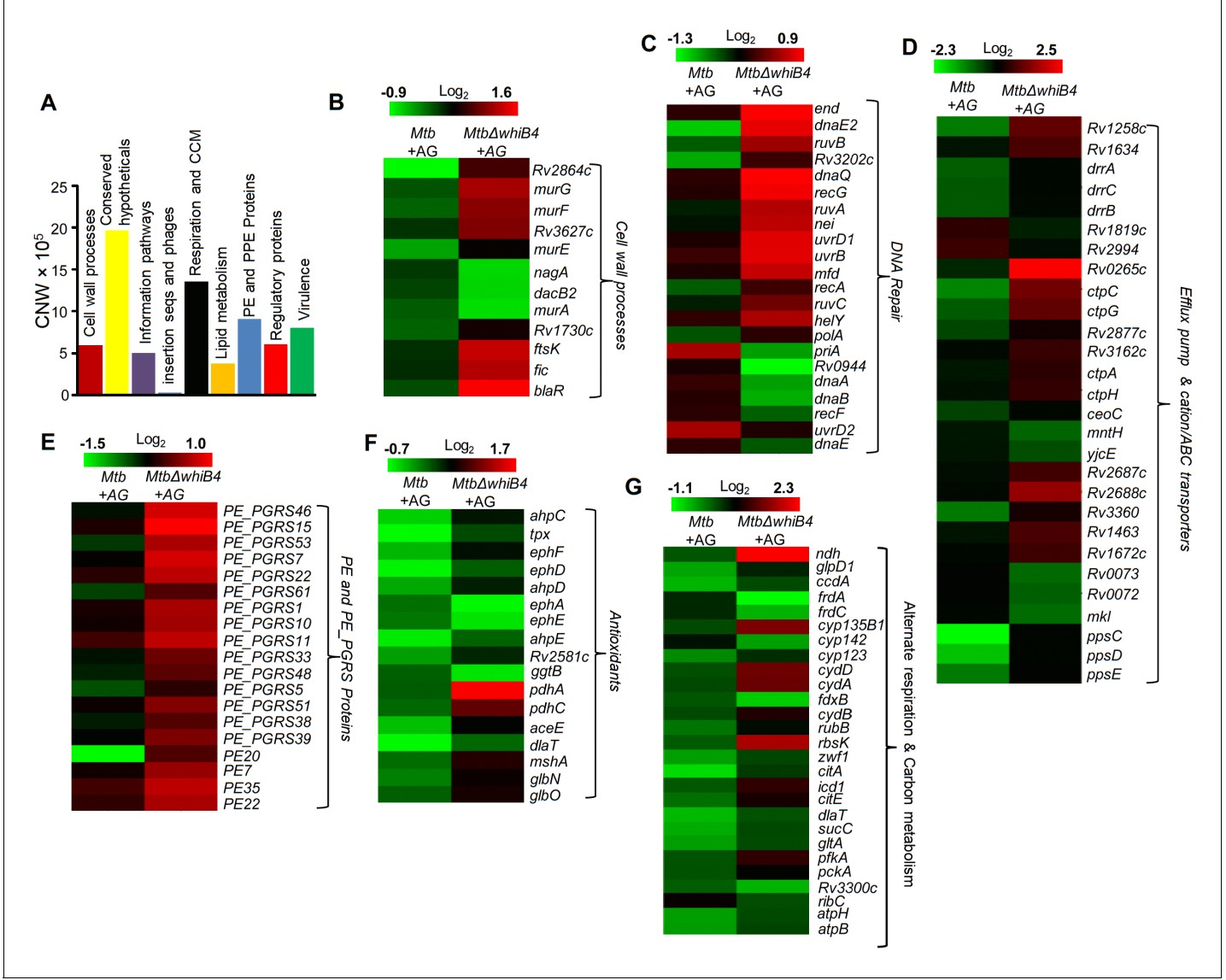

**Figure 6.** WhiB4 regulates response to AG in *Mtb*. (**A**) Cumulative node weight intensities (CNW) of different functional classes regulated by WhiB4 upon AG treatment. (**B–G**) Heat maps depicting expression of genes (log₂fold-change, p≤0.05) coordinating cell wall processes, alternate respiration and CCM, antioxidants, DNA repair, PE and PE_PGRS and drug efflux pumps in case of *Mtb* and *MtbΔwhiB4* treated with AG for 6 hr as described in Materials and methods. Cumulative node weight intensities for different functional classes are available in *Figure 6—source data 1*.

The following source data and figure supplements are available for figure 6:

**Source data 1.** Cumulative node weight intensities for different functional classes as depicted in *Figure 6A*.
**Figure supplement 1.** qRT-PCR analysis of *whiB4* expression in *MtbΔmshA*, *mshA-comp*, and *mshA-OE* strains.
**Figure supplement 2.** qRT-PCR analysis of *MtbΔwhiB4* exposed to 10X AG for 6 hr.
**Figure supplement 3.** Vancomycin-BODIPY staining of different *Mtb* strains.

macrophages using Mrx1-roGFP2 biosensor as described earlier. Under both culture conditions, *MtbΔwhiB4* robustly maintained intramycobacterial $E_{MSH}$, whereas Atc-induced overexpression of *whiB4* in *MtbΔwhiB4* showed a significant oxidative shift (*Figure 7*). In sum, our results suggest that

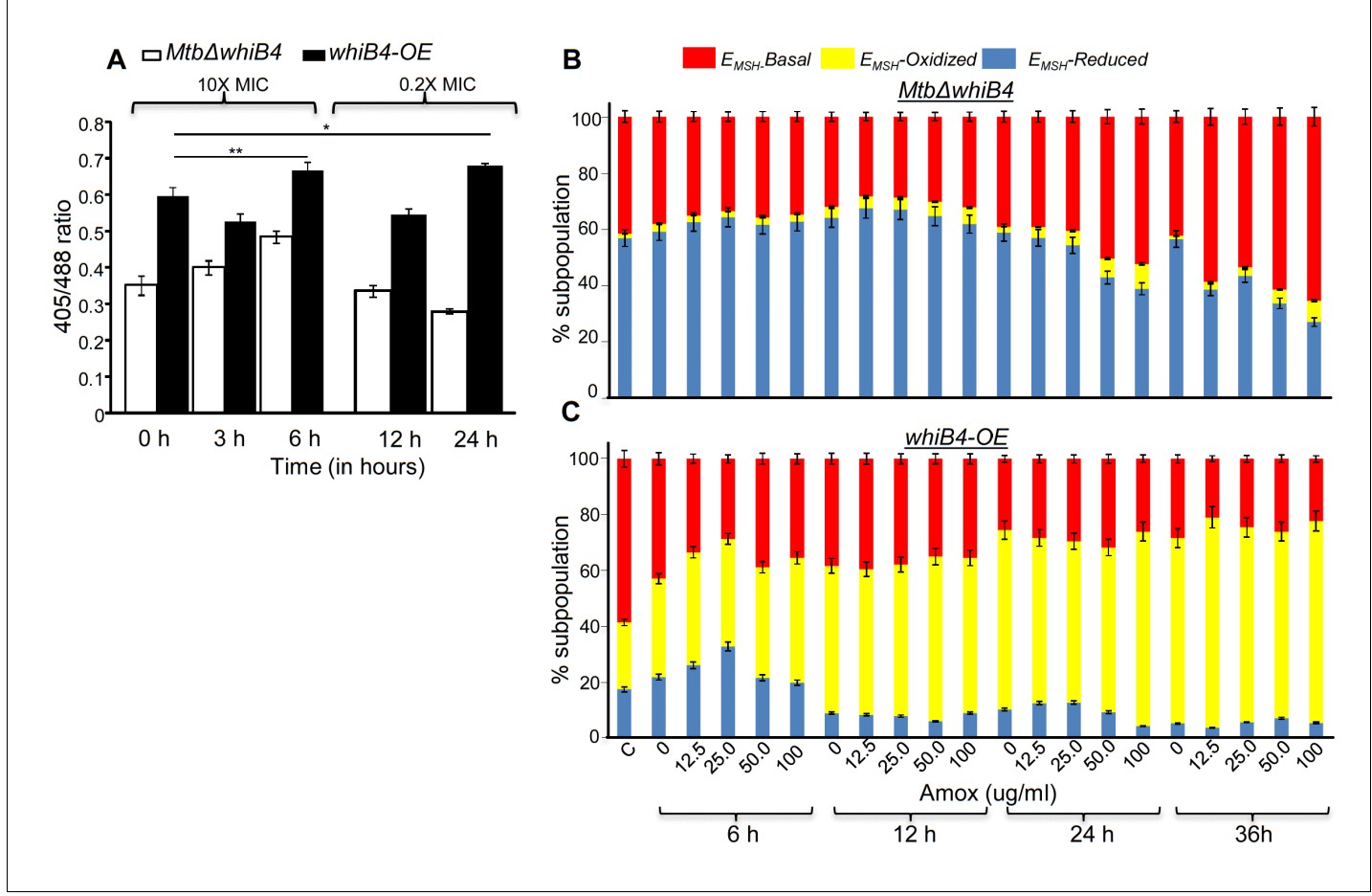

**Figure 7.** WhiB4 regulates AG-induced oxidative shift in $E_{MSH}$ of *Mtb* both in vitro and during infection. (A) *MtbΔwhiB4* and *whiB4-OE* expressing Mrx1-roGFP2 were treated with lethal (10X MIC) and sub-lethal (0.2 X MIC) concentrations of AG and ratiometric response was measured by flow cytometry at indicated time points. (B) PMA differentiated THP-1 cells were infected with *MtbΔwhiB4* and *whiB4-OE* expressing Mrx1-roGFP2 (MOI:10) and treated with indicated concentrations of Amox in the presence of Clav (8 µg/ml) as described in Materials and methods. At indicated time points, ~30,000 infected macrophages were analyzed by flow cytometry to quantify changes in *Mtb* subpopulations displaying variable $E_{MSH}$ as described in Materials and methods. *p≤0.05, **p≤0.01 and ***p≤0.001.

WhiB4 can mediate AG tolerance by regulating multiple mechanisms, including PG biogenesis, SOS response, and redox balance.

## *Mtb* WhiB4 regulates BlaC in a redox-dependent manner

Our data indicated that WhiB4 modulates the expression of genes involved in *β*-lactam tolerance (*blaR* and *blaC*) and redox metabolism (*mshA*, *ahpCD*, and *tpx*). Using qRT-PCR, we confirmed that the expression of *blaR* and *blaC* was 8.43 ± 4.75 and 2.23 ± 0.19 fold higher, respectively, in *MtbΔwhiB4* as compared to wt *Mtb* upon exposure to AG (*Figure 6—figure supplement 2*). Next, we examined WhiB4 interaction with upstream sequences of *blaR* and *blaC* using EMSA. Earlier, we have shown that WhiB4 contains a 4Fe-4S cluster, which is extremely sensitive to degradation by atmospheric oxygen (*Chawla et al., 2012*). Moreover, WhiB4 lacking the Fe-S cluster (apo-WhiB4) binds DNA and represses transcription upon oxidation of its cysteine thiols and formation of disulfide-linked oligomers, while reduction of disulfides reversed WhiB4 oligomerisation, DNA binding, and repressor function (*Chawla et al., 2012*). We generated thiol-reduced and -oxidized forms of apo-WhiB4 as described previously (*Chawla et al., 2012*). The oxidized and reduced apo-WhiB4 fractions were incubated with $^{32}$P-labeled DNA fragments of *blaC* (~100 bp upstream) and *blaR* (~180 bp upstream) and *blaC/blaR*-promoter complex formation was visualized using EMSA.

As shown in *Figure 8A and B*, oxidized apo-WhiB4 binds *blaC/blaR*-promoter DNA in a concentration-dependent manner, whereas this binding was significantly reversed in case of reduced apo-WhiB4. Since WhiB4 bind to its own promoter (*Chawla et al., 2012*), we confirmed that oxidized apo-WhiB4 binds to its promoter in concentrations comparable to that required for binding *blaC* and *blaR* upstream sequences (*Figure 8C*). We also performed competition assays using *blaC* and *blaR* upstream sequences as positive controls, while promoter fragment of *Rv0986* was utilized as a negative control. We found that 100-fold molar excess of *blaC* and *blaR* DNA fragments completely prevented apo-WhiB4 binding. However, the same concentration of an unlabeled *Rv0986* promoter fragment was inefficient to out-compete apo-WhiB4 association with *blaC* and *blaR* DNA fragments (*Figure 8—figure supplement 1*). Next, we performed in vitro transcription assays using a highly sensitive *Msm* RNA polymerase holoenzyme containing stoichiometric concentrations of principal Sigma factor, SigA (RNAP-$\sigma^A$) (*Chawla et al., 2012*) and determined the consequence of WhiB4 on *blaC* transcript. As shown in *Figure 8D*, addition of oxidized apo-WhiB4 noticeably inhibited transcription from *blaC* promoter, whereas reduced apo-WhiB4 restored normal levels of *blaC* transcript. Lastly, we directly measured BlaC activity in the cell-free extracts derived from wt *Mtb*, *MtbΔwhiB4*, and *whiB4-OE* strains using a chromogenic *β*-lactam nitrocefin as a substrate (*Flores et al., 2005b*). Cell-free extracts of *MtbΔwhiB4* possessed ~70% higher and *whiB4-OE* showed ~30% reduced

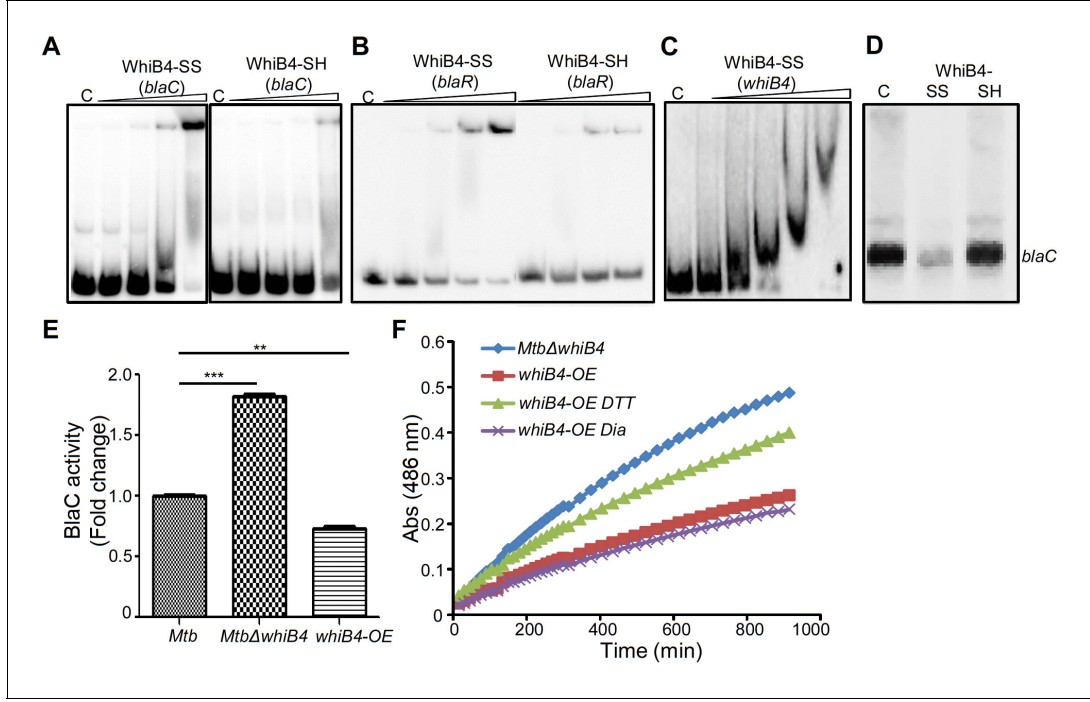

**Figure 8.** Regulation of *β*-lactamase by WhiB4 in a redox-dependent manner. Oxidized (WhiB4-SS) and reduced (WhiB4-SH) forms of apo-WhiB4 were prepared. The concentrations of apo-WhiB4 used for EMSAs were 0.5, 1, 2, and 4 µM. EMSA reactions were performed with 0.5 nM $^{32}$P-labelled *blaC* (A), *blaR* (B) and *whiB4* (C) promoter DNA fragments. C: DNA binding in the absence of WhiB4 in each panel. (D) Effect of WhiB4 on in vitro transcription. Single-round transcription assays show that RNAP-$\sigma^A$ efficiently directs transcription from the *blaC* promoter. 100 nM of *blaC* promoter DNA fragment was pre-incubated with either 1 µM WhiB4-SS or WhiB4-SH and subjected to transcription by RNAP-$\sigma^A$ as described in Materials and methods. C: *blaC* transcript in the absence of WhiB4. (E) 100 µg of cell-free lysates derived from exponentially grown (OD$_{600}$ of 0.6) wt *Mtb*, *MtbΔwhiB4* and *whiB4-OE* were used to hydrolyze nitrocefin. *β*-lactamase activity was measured by monitoring absorbance of hydrolyzed nitrocefin at 486 nm as described in Materials and methods. The fold change ratios clearly indicate a significantly higher or lower *β*-lactamase activity in *MtbΔwhiB4* or *whiB4-OE*, respectively, as compared to wt *Mtb*. p-Values are shown for each comparison. (F) *whiB4-OE* strain was pre-treated with 5 mM of DTT or Diamide and *β*-lactamase activity in cell-free lysates was compared to *MtbΔwhiB4* over time. *p≤0.05, **p≤0.01 and ***p≤0.001. Data are representative of at least two independent experiments done in duplicate.

The following figure supplement is available for figure 8:

**Figure supplement 1.** EMSA cold competition assay.

nitrocefin hydrolysis as compared to wt *Mtb*, respectively (*Figure 8E*). We have earlier shown that WhiB4 predominantly exists in an oxidized apo-form upon overexpression inside mycobacteria during aerobic growth (*Chawla et al., 2012*). Therefore, decreased BlaC activity upon WhiB4 overexpression is most likely a consequence of oxidized apo-WhiB4-mediated repression of *blaC in vivo*. To clarify the physiological relevance of redox- and *whiB4*-dependent transcription of *blaC*, we shifted the internal redox balance of *whiB4-OE* using a cell permeable thiol-oxidant, diamide (5 mM), or a thiol-reductant, DTT (5 mM), and measured nitrocefin hydrolysis by cell-free extracts. We have previously reported that treatment with 5 mM diamide or DTT did not adversely affect growth of *Mtb* (*Singh et al., 2009*). However, treatment with DTT significantly reduced disulfide-linked oligomers of oxidized apo-WhiB4 to regenerate WhiB4 thiols in vivo (*Chawla et al., 2012*). Pretreatment of *whiB4-OE* with DTT largely restored BlaC activity to *MtbΔwhiB4* levels, whereas diamide did not lead to further decrease in BlaC activity (*Figure 8F*). Effective reduction of disulfides in oxidized apo-WhiB4 by DTT may have led to loss of WhiB4 mediated DNA binding and transcriptional repression, thereby causing elevated *blaC* expression and activity in *whiB4-OE*. Taken together, these results led us to conclude that WhiB4 regulates *β*-lactamase expression and activity in a redox-dependent manner.

## WhiB4 regulates survival in response to β-lactams in *Mtb*

Based on the above results, we hypothesize that WhiB4-sufficient and -deficient strains would have differential susceptibility toward *β*-lactams. We found that *MtbΔwhiB4* uniformly displayed ~4–8 fold higher MICs against *β*-lactams as compared to wt *Mtb* (*Table 2*). This effect was specific to *β*-lactams, as the loss of WhiB4 did not alter MICs for other anti-TB drugs such as INH and RIF (*Table 2*). More-interestingly, over-expression of WhiB4 displayed ~2–4 fold greater sensitivity toward *β*-lactams as compared to wt *Mtb* (*Table 2*). We predicted that if WhiB4 is controlling tolerance to *β*-lactams by regulating *blaC* expression, we would see variations in inhibitory concentrations of Clav against wt *Mtb*, *MtbΔwhiB4*, and *whiB4-OE* at a fixed concentration of Amox. As expected, inhibition of *MtbΔwhiB4* by 10 µg/ml of Amox requires four fold and eight fold higher Clav as compared to wt *Mtb* and *whiB4-OE* strains, respectively (*Figure 9A*). Phenotypic data are in complete agreement with the higher and lower BlaC activity in *MtbΔwhiB4* and *whiB4-OE*, respectively. Studies in animals and humans have demonstrated higher efficacy of *β*-lactams and *β*-lactamase inhibitor combination against MDR/XDR-TB. Our results show that WhiB4 overexpression significantly elevated

**Table 2.** Minimum inhibitory concentrations (MICs) of cell wall targeting drugs for different *Mycobacterium tuberculosis* strains.

Source data file for the calculation of MIC values is available in *Table 2–source data 1*.

| Drugs | µg/ml | | |
| --- | --- | --- | --- |
| | *Mtb* | *MtbΔwhib4B4* | *whiB4-OE* |
| Amoxicillin | 80 | >160 | 40 |
| Ampicillin | 500 | 4000 | 250 |
| Cloxacillin | 400 | 800 | 200 |
| Carbenicillin | 1024 | 4096 | 512 |
| Meropenem | 5 | 20 | 2.5 |
| Penicillin | 200 | 800 | 100 |
| Lysozyme | 50 | 200 | 25 |
| Vancomycin | 10 | 80 | 2.5 |
| Isoniazid | 0.0625 | 0.0625 | 0.03125 |
| Rifampicin | 0.0625 | 0.0625 | 0.0625 |

Source data 1. Percentage growth inhibition values for *Mtb*, *MtbΔwhiB4* and *whiB4-OE* in presence of different drugs for calculation of minimum inhibitory concentration (MIC).

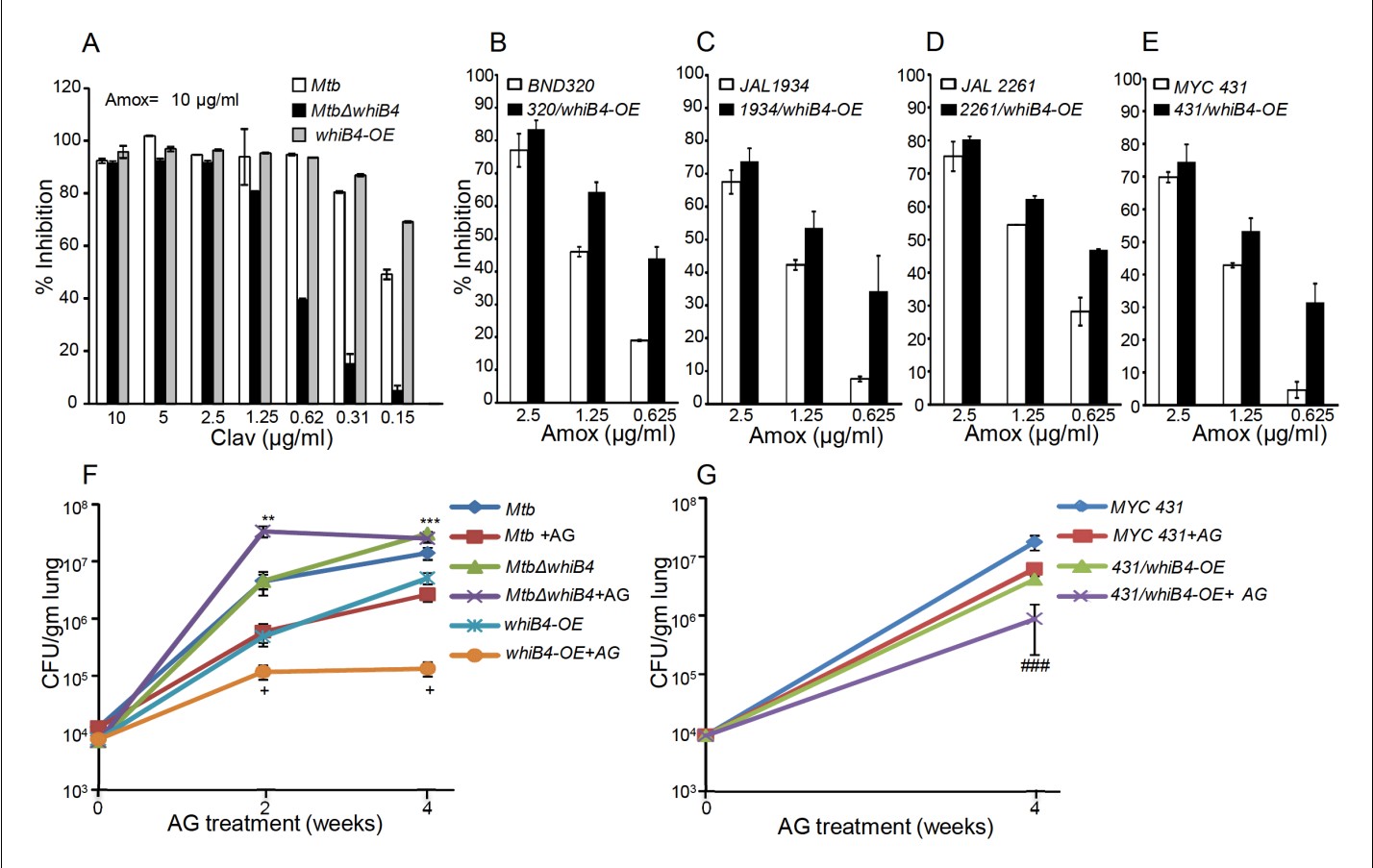

**Figure 9.** WhiB4 regulates AG tolerance in drug-sensitive and -resistant strains of *Mtb*. (A) Wt *Mtb*, *MtbΔwhiB4* and *whiB4-OE* were incubated with Amox (10 µg/ml) and different concentrations of Clav and % inhibition of growth was measured by AB assay as described in Materials and methods. To determine if WhiB4 modulates the sensitivity of AG in drug-resistant strains, WhiB4 was over-expressed in clinical strains (B) BND 320 (C) JAL 1934, (D) JAL 2261, and (E) MYC 431. Cells were incubated with Clav (8 µg/ml) and different concentrations of Amox. The percent growth inhibition was measured by AB assay as described in Materials and methods. WhiB4 modulates susceptibility to AG during acute infection in mice (F–G). Inbred BALB/c mice (n = 3) were given various strains of *Mtb* in the form of an aerosol and orally administered with Amox (200 mg/kg of body weight) and Clav (50 mg/kg of body weight) that is AG twice a day starting from day 3 post-infection. Bacterial burden in the lungs was assessed by checking the survival of *Mtb* strains using CFU analysis. Statistical significance for the pulmonic bacterial load was obtained as follows: by comparing the CFU obtained from AG-treated Wt *Mtb* and *MtbΔwhiB4* strains: **p≤0.01 and ***p≤0.001, by comparing CFU obtained from AG-treated Wt *Mtb* and *whiB4-OE* strains: + p≤0.05, by comparing CFU obtained from AG-treated MYC 431 and MYC 431/*whiB4*-OE strains: ### p≤0.001.

the capacity of *β*-lactams to inhibit drug-sensitive *Mtb*. To investigate whether WhiB4 overexpression similarly affects growth of drug-resistant strains, we over-expressed WhiB4 in clinical strains isolated from Indian patients (single-drug resistant [SDR; BND320], multi-drug resistant [MDR; JAL 2261 and JAL 1934] and extensively drug-resistant [XDR; MYC 431]) (*Bhaskar et al., 2014*; *Kumar et al., 2010*) and determined sensitivity toward Amox (at various concentrations) and Clav (8 µg/ml). As expected, drug-resistant strains over-expressing WhiB4 were ~2–4 fold more sensitive to Amox and Clav combinations than controls (*Figure 9B, C, D and E*).

Lastly, we asked if WhiB4 influences tolerance to AG during infection. Poor half-life of AG in mice makes it challenging to assess the efficacy of AG in vivo (*Rullas et al., 2015*). However, AG induces a marginal (~0.5 log reduction) killing of *Mtb* in an acute model for TB infection in mice (*Solapure et al., 2013*). Therefore, we compared bacillary load of *Mtb* strains in the lungs of mice during acute infection (see Materials and methods). Approximately 10⁴ bacterial cells were implanted into the lungs of BALB/c mice (*Figure 9F*) and at 3 days post-infection mice were treated with AG (200 mg/kg body weight of Amox and 50 mg/kg body weight of Clav) twice a day for 2 and

4 weeks. Bacterial numbers were determined in the infected lungs upon treatment. At 3 days post-infection, bacillary load was comparable between *Mtb* strains (*Figure 9F*). At 2 and 4 weeks post-treatment, wt *Mtb* exhibited ~7-fold reduction in bacillary load than untreated mice (*Figure 9F*). Overexpression of WhiB4 resulted in ~4- and~38-fold decline in CFU at 2 and 4 weeks, post-treatment, as compared to untreated animals (*Figure 9F*). In contrast, *MtbΔwhiB4* either displayed an increase (~8-fold) or maintained a comparable bacillary load at 2 or 4 weeks post-treatment, respectively, relative to untreated mice (*Figure 9F*). Lastly, we overexpressed WhiB4 in the MYC 431 XDR strain (431/*whiB4*-OE) and examined AG efficacy in mice as described earlier. As expected, WhiB4 overexpression increased the sensitivity of MYC 431 toward AG treatment at 4 weeks post-infection (*Figure 9G*). We documented that WhiB4 overexpression significantly affects the survival of *Mtb in vivo*, an outcome that is most likely due to WhiB4-directed repression of the antioxidant systems and *β*-lactamase. In conclusion, our results suggest that WhiB4 plays a central role in coordinating *Mtb* tolerance to AG.

## Discussion

We revealed a redox-based mechanism underlying tolerance to a *β*-lactam and *β*-lactamase inhibitor combination, which is actively considered to treat drug-resistant *Mtb* infections. Importantly, these findings should be viewed in light of recent studies debating the contribution of antibiotic-induced redox perturbations in antibiotic action and tolerance (*Kohanski et al., 2007*; *Foti et al., 2012*; *Brynildsen et al., 2013*; *Liu and Imlay, 2013*; *Keren et al., 2013*). We have shown how the primary targets of antibiotics (e.g. PG biogenesis and *β*-lactamase) and their secondary consequences (redox stress and metabolic perturbations) are functionally associated with each other through a redox-sensitive transcription factor, WhiB4, in *Mtb*. Considering the fact that drug-resistance in *Mtb* is a global burden, our results showing that WhiB4-mediated changes in redox potential of *Mtb* can potentiate killing of clinical drug-resistant forms of *Mtb* by AG are novel and unique. We identified the internal redox potential of *Mtb* as a crucial determinant of mycobacterial sensitivity to AG, and demonstrated the central role of WhiB4 in maintaining redox balance and regulating gene expression. Down-regulation of TCA cycle genes and up-regulation of the glyoxylate cycle in response to AG are consistent with the reports of elevated tolerance to diverse bactericidal antibiotics, including *β*-lactams, in bacteria with diminished fluxes through the TCA cycle (*Kohanski et al., 2007*; *Nguyen et al., 2011*). In concurrence with this, metabolomic profiling of *Mtb* in response to other anti-TB drugs elegantly showed that tolerance is accompanied with reduced TCA cycle activity and elevated fluxes through the glyoxylate shunt (*Nandakumar et al., 2014*). *Mtb* exhibits tolerance to antibiotics during non-replicating persistence in hypoxia (*Baek et al., 2011*). Under these conditions, drug tolerance was accompanied by a redirection of respiration from the energetically efficient route (e.g. NADH dehydrogenase I) to the less energy efficient course (e.g. NDH, CydAB oxidase), and any interference with this respiratory-switch over (e.g. CydAB mutation) leads to resensitization of mycobacteria to antibiotics (*Rao et al., 2008*; *Lu et al., 2015*). This seems to be a unifying theme underlying tolerance to conventional as well as the newly discovered anti-TB drugs bedaquiline (BDQ) and Q203 (*Lamprecht et al., 2016*). In support of this, we found that exposure of *Mtb* to AG elicited a transcriptional signature that indicated a shift from the energy efficient respiration to the energetically less favored pathways, as evidenced by a significant induction of *ndh* and *cydAB* transcripts and a down-regulation of *nuo*, *cydbc1*, and *atp A-H*. In bacteria, including *Mtb*, cytochrome *bd* oxidase also displays catalase and/or quinol oxidase activity (*Lu et al., 2015*; *Al-Attar et al., 2016*), which confers protection against oxidative stress and nitrosative stress. On this basis, upregulation of cytochrome *bd* oxidase in response to AG is indicative of oxidative stress in *Mtb*. Bactericidal antibiotics, including *β*-lactams, have been consistently shown to produce ROS as a maladaptive consequence of primary drug-target interaction on TCA cycle and respiration (*Kohanski et al., 2007*; *Lobritz et al., 2015*; *Dwyer et al., 2014*). While this proposal has been repeatedly questioned (*Liu and Imlay, 2013*; *Keren et al., 2013*), it is strongly reinforced by multiple independent studies demonstrating that tolerance to antibiotics is linked to the bacterial ability to nullify antibiotic-triggered ROS toxicity (*Nguyen et al., 2011*; *Wang and Zhao, 2009*; *Gusarov et al., 2009*; *Shatalin et al., 2011*). We confirmed that AG stimulates oxidative stress in *Mtb* in vitro and during infection. However, in contrast to other studies (*Kohanski et al., 2007*), oxidative stress was not associated with a breakdown of the NADH/NAD$^+$ homeostasis, likely reflecting

efficient ETC fluxes through NDH and cytochrome *bd* oxidase. In *Mtb*, rerouting of electron fluxes through cytochrome *bd* oxidase increases oxygen consumption (*Lamprecht et al., 2016*), which can trigger $O_2^{-\bullet}$ and $H_2O_2$ generation by univalent reduction of $O_2$ by the metal, flavin, and quinone containing cofactors of the respiratory enzymes (*Imlay, 2003, 2013*). Recently, it has been shown that intramycobacterial antioxidant buffer, MSH, protects *Mtb* from small molecule endogenous superoxide generators and ROS-generated by vitamin C (*Tyagi et al., 2015*; *Vilchèze et al., 2013*). Specific to AG, we found that anti-mycobactericidal activity is greatly potentiated in MSH-deficient mycobacterial strains, whereas a MSH overexpressing strain displayed tolerance. This is all consistent with the generation of ROS and MSH as key regulatory mechanisms underlying AG tolerance.

Studies indicated the importance of a broader range of physiological programs such as altered metabolic state and oxidative stress as contributory factors in antibiotic resistance. However, it is not clear if specific regulatory mediators exist which can assess physiological changes to regulate both primary drug targets and secondary consequences of drug-target interactions to functionally coordinate tolerance. Mechanisms of drug tolerance are either controlled by global changes in bacterial physiology by ppGpp or toxin-antitoxin (TA) modules (*Harms et al., 2016*). Further, regulatory systems such as SoxRS in other bacteria and WhiB7 in *Mtb* facilitate physiological changes required for formation of drug-tolerant persisters without specifically affecting the expression of direct targets of antibiotics (*Morris et al., 2005*; *Aly et al., 2015*). We, for the first time, identified WhiB4 as a transcriptional regulator of both the genetic determinants of β-lactam resistance (e.g. β-lactamase) and physiological changes associated with phenotypic drug tolerance in *Mtb* (e.g. redox balance).

Due to lack of an extracellular β-lactam-sensing domain in *Mtb* BlaR, how *Mtb* responds to β-lactam remains unknown. While several possibilities including the involvement of serine/threonine protein kinases (PknA/PknB) containing β-lactam interacting PASTA domains are suggested to regulate BlaR-BlaI activity (*Sala et al., 2009*; *Mir et al., 2011*), our findings implicate internal redox balance and WhiB4 in responding to β-lactams. We detected that oxidized apo-WhiB4 binds and represses the expression of BlaR and BlaC, whereas reduction reversed this effect. Loss of WhiB4 derepresses BlaR and stimulates the expression and activity of BlaC, possibly via BlaR-mediated cleavage of the repressor of *blaC* (i.e. BlaI). In addition to *blaC*, BlaI also binds to the promoters of genes encoding cytochrome *bd* oxidase and ATP synthase (*Sala et al., 2009*), both of which showed higher expression in *MtbΔwhiB4*. Altogether it indicates that regulatory function of BlaI is dependent upon the ability of WhiB4 to coordinate *blaR* expression in response to redox changes associated with β-lactam exposure. Our findings indicate a possible regulatory loop between the electron transport chain and β-lactam-induced oxidative stress where WhiB4/BlaI/BlaR may act as an important link between them (*Figure 10*). The biogenesis of PG is an energy requiring process and the cell wall damage caused by β-lactam antibiotics can perturb membrane function thereby affecting respiration, ATP generation, and redox balance. All these events can cause metabolic paralysis leading to inhibition of PG biogenesis and death. Supporting this notion, a recent study on the mechanisms of β-lactam toxicity showed that β-lactams cause metabolic instability due to activation of a futile cycle of PG biogenesis and degradation (*Cho et al., 2014*). Therefore, tolerance to β-lactams would require active cooperation between mechanisms to maintain metabolic function, redox balance and β-lactamase activity, which are partly regulated by WhiB4 in *Mtb*. Under unstressed conditions, uncontrolled expression of genes such as *blaC* and *cydAB* is prevented by WhiB4-mediated DNA binding and repression of the *blaR-blaI* locus. This is possible since the WhiB4 Fe-S cluster is uniquely sensitive to oxygen and a fraction of WhiB4 exists in the apo-oxidized form inside aerobically growing *Mtb* (*Chawla et al., 2012*). Since oxidized apo-WhiB4 is known to repress its own expression (*Chawla et al., 2012*), *Mtb* can down-regulate the expression of *whiB4* by elevating the levels of oxidized apo-WhiB4 in response to oxidative stress caused by β-lactams. The down-regulation of WhiB4 can reduce its negative influence on gene expression, necessary to adjust the expression of *blaI*, *blaR*, and *blaC* as well as genes involved in maintaining respiration and redox balance to neutralize β-lactam toxicity (*Figure 10*). Our data confirmed this by demonstrating consistent repression of *whiB4* expression by AG treatment, oxidative stress, and upon MSH loss (in *MtbΔmshA*). The down-regulation of *whiB4* in *MtbΔmshA* is most likely a compensatory strategy to tolerate AG in the absence of MSH, albeit unsuccessfully, indicating that both WhiB4 and MSH are together required to tolerate β-lactam antibiotics in *Mtb*. Furthermore, induction of *mshA* in *MtbΔwhiB4* in response to AG indicates a redox-regulatory loop between WhiB4 and MSH to tolerate oxidative and antibiotic stress in *Mtb*. Lastly, considering that WhiB4 might affect gene expression by altering nucleoid

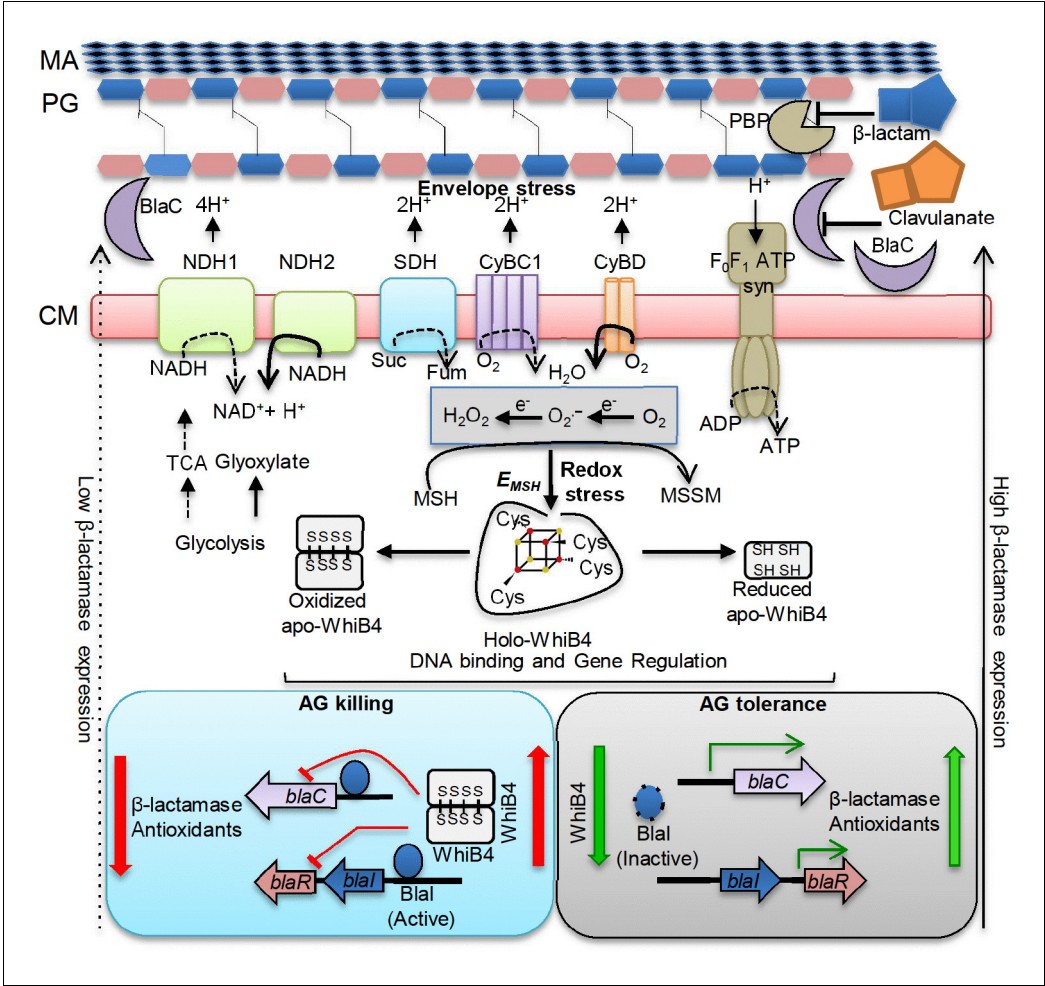

**Figure 10.** Model showing redox basis of AG tolerance in *Mtb*. Cell wall damage caused by AG can perturb the membrane integrity thereby affecting respiratory chain, redox balance, and ATP generation. All of this results in metabolic instability and AG-induced killing. To tolerate AG, *Mtb* redirects respiration from the energetically efficient route (e.g. NDH1, CyBC1) to the energetically poor one (e.g. NDH2, CyBD), and carbon metabolism from the TCA cycle to glyoxylate, glycolysis and gluconeogenesis. Rerouting of electron flux through CyBD can trigger generation of ROS ($O_2^{-\bullet}$ and $H_2O_2$) by univalent reduction of $O_2$ via metal-, flavin-, and quinone-containing respiratory enzymes. The intramycobacterial redox buffer, MSH, detoxifies ROS to protect *Mtb* from AG. The oxidative shift in $E_{MSH}$ of *Mtb* in response to AG serves as a cue to calibrate the expression of β-lactamase, PG enzymes, carbon metabolism, antioxidants, and alternate respiration via WhiB4. Under native conditions, $O_2$-induced loss of WhiB4 Fe-S cluster generates oxidized apo-WhiB4, which binds and represses the expression of *blaR* and *blaC*. Reduction of oxidized apo-WhiB4 disulfides reversed this effect. Down-regulation of *whiB4* in response to AG derepresses *blaR* and stimulates expression of *blaC* directly and/or indirectly via BlaR-mediated cleavage of the *blaC* repressor (i.e. BlaI) to induce AG tolerance. Accumulation of oxidized apo-WhiB4 upon overexpression led to hyper-repression of BlaC activity and oxidative shift in $E_{MSH}$ to potentiate mycobactericidal activity of AG. Since genes associated with alternate-respiration (e.g. CyBD) and energy metabolism (e.g. ATP synthase) are also regulated by BlaI, our results suggest cross-talk between WhiB4 and BlaI pathways resulting in AG tolerance of *Mtb*. Altogether, WhiB4 couples the changes in the redox physiology of *Mtb* triggered by AG to the expression of genes involved in antibiotic tolerance and redox homeostasis. MA: Mycolic acid, CM: Cytoplasmic membrane, NDH1: NADH-dehydrogenase I (*nuo* operon), NDH2: NADH dehydrogenase 2 (*ndh*), CyBD: Cytochrome BD oxidase, CyBC1: Cytochrome BC1-aa3 oxidase, $F_0F_1$ ATP syn: ATP Synthase, PBP: Penicillin-binding proteins and SDH: Succinate Dehydrogenase. Bold or dashed arrows indicate increased or decreased electron flow through respiratory complexes, respectively, based on gene expression data.

architecture (*Chawla et al., 2012*), the exact details of how WhiB4 regulates global gene expression is a part of an ongoing study. Data from this genome-scale DNA binding study (ChIP-seq) indicate that WhiB4 binds largely in a non-specific fashion to the *Mtb* chromosome with a particular preference to GC-rich regions including an intergenic region of *blaI-blaR* (>70% GC-rich) (manuscript in preparation).

In summary, our study discovered a new redox-based mechanism of AG tolerance in *Mtb*. In particular, WhiB4 functions as an important regulatory protein that integrates internal redox changes triggered by $\beta$-lactams to fine-tune the expression of both genetic and phenotypic determinants of antibiotic tolerance in *Mtb*. Based on this work, we predict that compounds/drugs targeting bacterial systems that remediate oxidative damage (e.g. 4-butyl-4-hydroxy-1- (4-hydroxyphenyl)−2-phenylpyrazolidine-3,5-dione) (*Gold et al., 2012*), elevate endogenous ROS (e.g. clofazimine/vitamin C) (*Bhaskar et al., 2014*; *Vilchèze et al., 2013*), inhibit respiration (e.g. Q2O3) (*Lamprecht et al., 2016*), and block ATP homeostasis (e.g. bedaquiline) (*Lamprecht et al., 2016*) could be be effective companions to potentiate the action of $\beta$-lactam and $\beta$-lactamase combinations in *Mtb*.

# Materials and methods

## Bacterial strains, mammalian cells, and growth conditions

Details of mycobacterial strains and reagents used in this study are given in *Supplementary file 3A* and *Supplementary file 3B*. The mycobacterial strains were grown aerobically in 7H9 broth or 7H11 agar supplemented with 0.2% glycerol, Middlebrook Oleic acid Albumin Dextrose-Catalase (OADC) or 1X Albumin Dextrose Saline (ADS) enrichment and 0.1% Tween 80 (broth). *E. coli* cultures were grown in LB medium. Antibiotics were added as described earlier (*Chawla et al., 2012*). For WhiB4 overexpression, *whiB4-OE* strain was grown aerobically to an $OD_{600}$ of 0.3, followed by induction with 200 ng/ml anhydrotetracycline (Atc) at 37°C for 18 hr. The human monocytic cell line THP-1 (RRID:CVCL_0006) was differentiated using 10–15 ng/ml phorbol 12-myristate 13-acetate (PMA) and cultivated for infection experiments as described previously (*Padiadpu et al., 2016*). THP-1 (ATCC TIB-202) cells authenticated by STR analysis by ATCC were treated with 25 µg/ml of Plasmocin for 3 weeks and tested negative for mycoplasma contamination by *DE-MyoX* Mycoplasma PCR Detection Kit.

## Drug sensitivity assay

Sensitivity to various drugs was determined using the microplate alamar blue assay (AB). AB assay was performed in 96-well flat bottom plates. *Mtb* or *Msm* strains were cultured in 7H9-ADS medium and grown till exponential phase ($OD_{600}$ of 0.6). Approximately $1 \times 10^5$ bacteria were taken per well in a total volume of 200 µl of 7H9-ADS medium. Wells containing no *Mtb* were used for autofluorescence control. Additional controls consisted of wells containing cells and medium only. Plates were incubated for 5 days (*Mtb*) or 16 hr (*Msm*) at 37°C, 30 µl (0.02% wt/vol stock solution) Alamar blue was added. Plates were re-incubated for color transformation (blue to pink). Fluorescence intensity was measured in a SpectraMax M3 plate reader (Molecular Device) in top-reading mode with excitation at 530 nm and emission at 590 nm. Percentage inhibition was calculated based on the relative fluorescence units and the minimum concentration that resulted in at least 90% inhibition was identified as MIC.

## Intracellular superoxide detection

*Mycobacterium bovis BCG* was cultured in 5 mL of Middlebrook 7H9 medium with 10% albumin-dextrose-saline (ADS) supplement at 37°C and grown till $OD_{600}$ of 0.4. The cultured bacteria were centrifuged to aspirate out the medium and re-suspended with fresh 7H9 medium. This bacterial solution was incubated with AG for 3 hr and 6 hr time points and 100 µM DHE was added for 1 hr in dark. The suspension was centrifuged to aspirate out any excess compounds and DHE in the medium. The collected bacterial pellet was re-suspended with acetonitrile and the cells were lysed using a probe sonicator for 3 min on ice. The cell lysate was then removed by centrifugation and the supernatant was separated and injected in Agilent high-performance liquid chromatograph (HPLC) attached with a fluorescence detector (excitation at 356 nm; emission at 590 nm) for analysis. Zorbax SB C-18 reversed-phase column (250 × 4.6 mm, 5 µm) was used and water: acetonitrile (0.1%

trifluoroacetic acid) was applied as mobile phase while flow rate was maintained at 0.5 ml/min. The HPLC method used was as described previously (*Kalyanaraman et al., 2014*).

## Intracellular NADH/ NAD⁺ ratio

NADH/NAD$^+$ ratios upon AG treatment were determined by NAD$^+$/NADH Quantification Kit (Sigma-Aldrich, USA). *Mtb* cells were cultured to OD$_{600}$ of 0.4 and treated with Amox-Clav combination for various time points (6, 12, and 24 hr). Ten milliliter of culture was harvested and washed with 1X PBS and NADH/NAD$^+$ ratio was determined according to the manufacturer's instructions.

## Detection of intracellular ROS

ROS generation upon AG treatment was assessed using a peroxide detection agent, 5-(and 6)-chloromethyl-2', 7'-dichlorodihydrofluorescein diacetate, acetyl ester (CM-H2DCFDA; Invitrogen USA, ThermoFisher Scientific). The reagent is converted to a fluorescent product by cellular peroxides/ROS as determined by flow cytometry. *Mtb* cells were cultured to mid-logarithmic phase (OD$_{600}$ of 0.4), and AG treatment was given for 3 hr and 6 hr. At each time point, 500 µL of culture was aliquoted and incubated with 20 µM of CM-H2DCFDA in dark (30 min) at 37°C. Cells were washed with 1X PBS and analyzed by FACS Verse flow cytometer (BD Biosciences, San Jose, CA). CM-H2DCFDA fluorescence was determined (excitation at 488 nm and emission at 530 nm) by measuring 10,000 events/sample.

## Nitrocefin hydrolysis assay

$\beta$-lactamase activity in *Mtb* strains was determined using a spectrophotometer by hydrolysis of nitrocefin, a chromogenic cephalosporin substrate that contains a $\beta$-lactam ring. Bacterial cultures were grown to an OD$_{600}$ of 0.6–0.8, and cells were harvested and lysed using bead beater (FastPrep Instrument, MP Bio). The cell-free lysate was clarified by centrifugation and 100 µg of lysate was incubated with 100 µM Nitrocefin. Hydrolysis of nitrocefin was monitored at 486 nm using a SpectraMax M3 plate reader (Molecular Devices) at regular intervals. Fold activity was calculated based on changes in absorbance at 486 nm over time. Normalization was performed by Bradford estimation of total protein in the cell-free lysates.

## Microarray hybridization and data analysis

For microarray analyses, wt *Mtb* and *MtbΔwhiB4* strains were cultured to an OD$_{600}$ of 0.4 and exposed to 1X (10 µg/ml of Amox and 8 µg/ml of Clav), 5X (50 µg/ml of Amox and 8 µg/ml of Clav) and 10X (100 µg/ml of Amox and 8 µg/ml of Clav) MIC of AG for 6 hr or 12 hr. For CHP stress, wt *Mtb* grown similarly was treated with 250 µM of CHP for 2 hr and samples was processed for microarrays. Total RNA was isolated from samples (taken in replicates), processed and hybridized to *Mtb* Whole Genome Gene Expression Profiling microarray- G2509F (AMADID: G2509F_034585, Agilent Technologies PLC) and data were analyzed as described (*Mehta et al., 2016*). DNA microarrays were provided by the University of Delhi, South Campus, MicroArray Centre (UDSC-MAC). RNA amplification, cDNA labeling, microarray hybridization, scanning, and data analysis were performed at the UDSC-MAC as described (*Mehta et al., 2016*). Slides were scanned on a microarray scanner (Agilent Technologies) and analyzed using GeneSpring software. Results were analyzed in MeV and considered significant at p≤0.05. The normalized data from the microarray gene expression experiment have been submitted to the NCBI Gene Expression Omnibus and can be queried via Gene Expression Omnibus series accession number GSE93091 (AG exposure) and GSE73877 (CHP exposure).

## Constructing AG response network of *Mtb*

Global PPI network was generated using the dataset described in the previous studies (*Szklarczyk et al., 2015*; *Balázsi et al., 2008*; *Cui et al., 2009*; *Wang et al., 2010*; *Zeng et al., 2012*; *Ghosh et al., 2013*; *Turkarslan et al., 2015*). The RRIDs of two of these networks are SCR_005223 (STRING) and SCR_003167 (Database of interacting proteins based on homology). After constructing the global PPI network of *Mtb*, we then extracted those interactions that are specific for genes present in our transcriptome data. Our microarray-specific network consists of 34035 edges and 4016 nodes. The expression data was used for assigning weights to nodes and edge in

the PPI network to make it condition-specific. The formalism of node and edge weight calculation is given below.

Node weight: We calculated node weight (NW) values for each node in the network by multiplying the normalized intensity values with the corresponding fold-change (FC) values. These values were uniformly scaled by multiplying with $10^4$.

$$NW_i = FC_i \text{ x Normalized signal intensity}$$

where i denotes the node in the network.

Edge weight: In order to calculate the edge weight values, we first calculated Edge-betweenness (EB) using NetworkX, a python package (https://networkx.github.io/). The other link for the codes is; https://github.com/networkx/networkx, where the users can directly pull the codes for usage. The GitHub link for Zen library used for computing shortest paths in the network is; https://github.com/networkdynamics/zenlib/tree/master/src/zen.

These values were scaled by multiplying with $10^6$. The node weight values were used to calculate the edge weight (EW) values as follows: -

$$EW = EB \text{ x } \sqrt{NW_i \text{ x } NW_j}$$

where i and j denotes nodes present in an edge.

$$Edge\,cost = 1/EW$$

The main focus of the study was to identify the key players involved in regulating the variations in different conditions. We carried out shortest path analysis on the condition-specific networks and selected the paths that are most perturbed in these conditions. We implemented shortest path algorithm to obtain the results.

## Shortest path analysis

The edge cost values were used as an input for calculating all vs. all shortest paths in each condition using Zen (http://www.networkdynamics.org/static/zen/html/api/algorithms/shortest_path.html). More than 9,000,000 paths were obtained for each condition. In order to analyze the more significant paths, we ordered the paths on the basis of their path scores. Path score is the summation of the edge cost that constitutes a path. Based on the formula considered for calculating edge cost, lower path score indicates that the nodes in the path have higher expression. So, instead of analyzing 9,000,000 paths, we considered subnetworks, which comprise of top 1% of the network. These networks were visualized using Cytoscape 3 (*Shannon et al., 2003*). Our response networks competently explain the perturbations in the system upon exposure to different situations such as AG treatment and/or disruption of whiB4. The networks were further co-related to graph-theory-based methods and differentially regulated paths were recognized in each condition to construct sub-network for each condition (*supplementary file 3C*).

## qRT-PCR analysis

Total RNA was isolated as described previously (*Chawla et al., 2012*) and cDNA was synthesized (after DNase treatment) from 500 ng isolated RNA. Random oligonucleotide primers were used with iScript Select cDNA Synthesis Kit for cDNA synthesis. Gene-specific primers (*supplementary file 3D*) were selected for RT-PCR (CFX96 RT-PCR system, Bio-Rad) and iQ SYBR Green Supermix was used for gene expression analysis. In order to obtain meticulous expression levels, PCR expression was normalized and CFX Manager software (Bio-Rad) was utilized for data analysis. Gene expression was normalized to *Mtb* 16S rRNA expression.

## Electrophoretic mobility shift assays (EMSA)

The histidine-tagged WhiB4 purification and generation of reduced or oxidized apo-WhiB4 was done as described previously (*Chawla et al., 2012*). For EMSA assays, the promoter fragments of *whiB4, blaC,* and *blaR* (~100–180 bp upstream of translational start codon) were PCR amplified from the *Mtb* genome and the 5' end was labeled using [γ-$^{32}$P]-ATP labeled oligonucleotides by using T4 polynucleotide kinase (MBI Fermentas) as per the manufacturer's instructions (*supplementary file*

*3D*). Binding reactions were performed in 1X TBE buffer (100 mM Tris, 90 mM boric acid and 1 mM EDTA; pH 8.33) for 30 min and 5% polyacrylamide gel was used to resolve protein-DNA complexes. For competition with unlabeled DNA, fragments of *blaC, blaR,* and *Rv0986* (~100–180 bp upstream of translational start codon) were PCR amplified from the *Mtb* genome and used in various amounts to outcompete binding of oxidized apo-WhiB4 to $^{32}$P-labeled DNA fragments. Gels were exposed to auto radiographic film and visualized via phosphoimaging (GE).

## In vitro transcription assays

50 nM of DNA fragment containing the *blaC* promoter and apo-WhiB4 (oxidized or reduced) were incubated in transcription buffer; 50 mM Tris HCl, (pH 8.0), 10 mM magnesium acetate, 100 μM EDTA, 100 μM DTT, 50 mM KCl, 50 μg/ml BSA, and 5% glycerol) for 30 min at room temperature. Single-round transcription assay was initiated by the addition of *Msm* RNAP-σ$^A$ holo enzyme (100 nM), 100 μM NTPs and 1 μCi α-$^{32}$P-UTP and incubated at 37°C for 20 min. Reactions were terminated with 2X stop dye (95% formamide, 0.025% (w/v) bromophenol blue, 0.025% (w/v) xylene cyanol, 5 mM EDTA and 0.025% SDS and 8 M urea) and heated at 95°C for 5 min followed by snap chilling in ice for 2 min. The transcripts were resolved by loading samples on to 6% urea-PAGE.

## Intramycobacterial E$_{MSH}$ measurement in vitro and during infection

Mycobacterial strains expressing Mrx1-roGFP2 were grown in 7H9 medium to OD$_{600}$ of 0.4 and exposed to various concentrations of AG. For measurements, cells were treated with 10 mM N-Ethylmaleimide (NEM) for 5 min at room temperature (RT) followed by fixation with 4% paraformaldehyde (PFA) for 15 min at RT. After washing three times with 1X PBS, bacilli were analyzed using BD FACS Verse Flow cytometer (BD Biosciences). The biosensor response was measured by analyzing the ratio at a fixed emission (510/10 nm) after excitation at 405 and 488 nm as described (*Bhaskar et al., 2014*). Data were analyzed using the FACSuite software. For measuring intramycobacterial E$_{MSH}$ during infection, THP-1 cells were treated with 15 ng/ml phorbol 12-myristate 13-acetate (PMA) for 20 hr to differentiate them into macrophages. Differentiated cells were then allowed to rest for 2 days, to ensure a resting phenotype before infection. PMA-differentiated THP-1 cells were infected with *Mtb* strains expressing Mrx1-roGFP2 at a multiplicity of infection (MOI) of 10 for 4 hr at 37°C. After 4 hr of infection, cells were washed with pre-warmed RPMI and amikacin treatment (0.2 mg/ml for 2 hr) was given to remove extracellular bacteria. Subsequently cells were washed and resuspended in fresh RPMI media containing various concentrations of Amox (12.5, 25, 50, and 100 μg/ml of Amox) and Clav (8 μg/ml) for 6, 12, 24, and 36 hr. At the indicated time points, infected macrophages were treated with NEM/PFA, washed with 1X PBS, and analyzed by flow cytometry as described previously (*Padiadpu et al., 2016*).

## Survival Assay upon AG treatment in vitro and during infection

*Mtb* strains were grown aerobically to OD$_{600}$ of 0.4, followed by treatment with various concentrations of AG. At defined time-points post-exposure, cells were serially diluted and plated on OADC-7H11 agar medium for enumerating CFUs. To determine the effect of AG during infection, ~20,000 THP-1 cells (PMA differentiated) were infected with wt *Mtb* in a 96-well plate (MOI:10) as described earlier (*Padiadpu et al., 2016*). Briefly, THP-1 monocytes were treated with 15 ng/ml of PMA for 20 hr to differentiate them into macrophages. Differentiated cells were then allowed to rest for 2 days and infected with *Mtb* H37Rv expressing Mrx1-roGFP2 at a MOI of 10 for 4 hr at 37°C. After 4 hr of infection, cells were washed with pre-warmed RPMI and amikacin treatment (0.2 mg/ml for 2 hr) was given to remove extracellular bacteria. Subsequently, cells were washed, fresh RPMI media was added and infected macrophages were exposed to AG (100 μg/ml of Amox and 8 μg/ml of Clav). At various time points, macrophages were lysed using 0.06% SDS-7H9 medium and released bacteria were serially diluted and plated on OADC-7H11 agar medium for CFU determination.

## Vancomycin-BODIPY staining

The pattern of nascent PG synthesis was observed by fluorescent staining as described (*Thanky et al., 2007*). *Mtb* strains were grown to exponential phase (OD$_{600}$ 0.6) in 7H9 medium. 1 ml of culture was incubated with 1 μg/ml Vancomycin-BODIPY (BODIPY FL Vancomycin) for 16 hr under standard growth conditions. Cells were pelleted to remove excess stain and fixed with PFA.

After washing with 1X PBS, culture aliquots (20 µl) were spread on slides and allowed to air dry. The bacterial cells were visualized for BODIPY FL Vancomycin fluorescence (excitation at 560 nm and emission at 590 nm) in a Leica TCS Sp5 confocal microscope under a 63X oil immersion objective. Staining pattern of more than 150 cells was observed for each strain and cell length was measured using Image J software.

## Aerosol infection of mice

For the acute model of infection, BALB/c mice were infected by aerosol with 10,000 bacilli per mouse with the *Mtb H37Rv, MtbΔwhiB4, whiB4-OE,* MYC 431, and MYC 431/*whiB4-OE* strains as described previously (*Solapure et al., 2013*). For assured over-expression of WhiB4, doxycycline (1 mg/ml in 5% sucrose solution) was supplied in drinking water. Dosages of Amox and Clav were maintained at 200 mg/kg of body weight and as 50 mg/kg of body weight, respectively, and the drugs were administered orally twice a day. At specific time points, mice were sacrificed and their lungs were removed and processed for investigation of bacillary load. CFUs were determined by plating appropriate serial dilutions on 7H11(supplemented with OADC) plates. Colonies were observed and counted after 3–4 weeks of incubation at 37°C.

## Statistical analysis

Statistical analyses were performed using the GraphPad Prism software (RRID: SCR_002798). The statistical significance of the differences between experimental groups (and controls where appropriate) was determined by two-tailed, unpaired Student's t test. Differences with a p value of $\leq 0.05$ were considered significant.

## Ethics statement

This study was carried out in strict accordance with the guidelines provided by the Committee for the Purpose of Control and Supervision on Experiments on Animals (CPCSEA), Government of India. The protocol was approved by the Committee on the Ethics of Animal Experiments of the International Centre for Genetic Engineering and Biotechnology (ICGEB), New Delhi, India (Approval number: ICGEB/AH/2011/2/IMM-26). All efforts were made to minimize the suffering.

## Acknowledgements

We are thankful to the University of Delhi South Campus MicroArray Centre (UDSCMAC), New Delhi for conducting microarray experiment. We are grateful to Dr. Harinath Chakrapani (IISER, Pune), Vasista Adiga (IISc, Bangalore) and Dr. Santosh Podder (IISc, Bangalore) for excellent technical help with DHE assay, FACS analysis and confocal microscopy, respectively. We thank Dr. William R. Jacobs, Jr. (Albert Einstein College of Medicine) for the *MtbΔmshA* and *mshA* complemented strains; David R. Sherman (Seattle Biomed, USA) for TetRO based *E.coli-mycobacterial* shuttle vector, *pEXCF-whiB4*; Dr. Y Av-Gay (University of British Columbia, Vancouver, British Columbia, Canada) for *MsmΔmshA* and *MsmΔmshD* mutants; Dr. Robert Husson (Children's Hospital Boston and Harvard Medical School, Boston, Massachusetts, United States of America) for *MsmΔsigH* mutant. The *Mtb* work was supported by the Wellcome-DBT India Alliance grant, WT-DBT/500034-Z-09-Z and IA/S/16/2/502700 (AS), and in part by Department of Biotechnology (DBT) Grant BT/PR5020/MED/29/1454/2012 (AS) and DBT-IISc program. AS is a Wellcome DBT India Alliance Intermediate Fellow. We acknowledge DBT-IISc-supported BSL3 facility for carrying out experiments on *Mtb* strains. We are thankful to Dr Vinay K Nandicoori for critical reading of the manuscript.

## Additional information

### Funding

| Funder | Grant reference number | Author |
| --- | --- | --- |
| Department of Biotechnology, Ministry of Science and Technology | BT/PR5020/MED/29/1454/2012 | Amit Singh |
| Wellcome | WT-DBT/500034-Z-09-Z | Amit Singh |

The funders had no role in study design, data collection and interpretation, or the decision to submit the work for publication.

## Author contributions

SM, Data curation, Formal analysis, Investigation, Methodology, Writing—original draft, Writing—review and editing; PS, Data curation, Formal analysis, Investigation, Methodology, Writing—review and editing; AB, KA, Data curation, Formal analysis, Investigation; PB, Data curation, Formal analysis, Investigation, Methodology; RKJ, Formal analysis, Investigation, Methodology; AM, Investigation, Methodology; RSR, AS, Conceptualization, Resources, Formal analysis, Supervision, Funding acquisition, Investigation, Methodology, Writing—original draft, Project administration, Writing—review and editing; VN, Formal analysis, Supervision, Investigation; NC, Supervision

## Author ORCIDs

Saurabh Mishra, http://orcid.org/0000-0003-1590-884X
Amit Singh, http://orcid.org/0000-0001-6761-1664

## Ethics

Animal experimentation: This study was carried out in strict accordance with the guidelines provided by the Committee for the Purpose of Control and Supervision on Experiments on Animals (CPCSEA), Government of India. The protocol was approved by the Committee on the Ethics of Animal Experiments of the International Centre for Genetic Engineering and Biotechnology (ICGEB), New Delhi, India (Approval number: ICGEB/AH/2011/2/IMM-26). All efforts were made to minimize the suffering.

# Additional files

## Supplementary files

• Supplementary file 1. Network analysis of *Mtb* in response to AG and CHP. (A) Raw microarray data sheet depicting log intensity values and absolute fold change values for Wt *Mtb*-treated 10XAG vs Wt *Mtb*-Untreated. (B) List of nodes present in top 1% network of for Wt *Mtb*-treated 10XAG vs Wt *Mtb*-Untreated. (C) List of Nodes common in top 1% network of Wt *Mtb*-treated 10XAG and Wt *Mtb* CHP treatments. (D) Raw microarray data sheet depicting log intensity values and absolute fold change values for Wt *Mtb* treated 1XAG and 5XAG vs Wt *Mtb* Untreated.

• Supplementary file 2. Global expression analysis of *Mtb* and *MtbΔwhiB4* in response to AG. (A) Raw microarray data sheet depicting log intensity values and absolute fold change values for *MtbΔwhiB4*-treated 10XAG vs Wt *Mtb*-treated 10XAG. (B) Raw microarray data sheet depicting log intensity values and absolute fold change values for *MtbΔwhiB4* treated 10XAG vs *MtbΔwhiB4* Untreated.

• Supplementary file 3. Description of strains, plasmids, chemicals , primers and details of top 1% network in response to AG. (A) List of bacterial strains and plasmids used in this study. (B) List of chemicals used in this study. (C) Network statistics for top 1% network. (D) Sequences of primers used in this study.

## Major datasets

The following datasets were generated:

| Author(s) | Year | Dataset title | Dataset URL | Database, license, and accessibility information |
| --- | --- | --- | --- | --- |
| Mishra S, Singh A | 2017 | Transcriptomic analysis of Mtb H37Rv and MtbΔwhiB4 on treatment to 10X Augmentin (AG) i.e. combination of 100 µg/ml of Amoxicillin and 8 µg/ml of | https://www.ncbi.nlm.nih.gov/geo/query/acc.cgi?acc=GSE93091 | Publicly available at the NCBI Gene Expression Omnibus (accession no: GSE93091) |

| | | | | | |
|---|---|---|---|---|---|
| | | | Clavulanate and lower Augmentin doses at 1X and 5X | | |
| Parikh P, Chawla M, Singh A | 2017 | Transcriptomic analysis of Mtb H37Rrv and MtbΔwhiB4 upon treatment with 0.25 mM CHP | https://www.ncbi.nlm.nih.gov/geo/query/acc.cgi?acc=GSE73877 | | Publicly available at the NCBI Gene Expression Omnibus (accession no: GSE73877) |

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

# Appendix 1

## Importance of generating AG-response network of *Mtb*

To assess the response of *Mtb* towards $\beta$-lactam and $\beta$-lactamase inhibitor combination(s), we generated transcriptome of *Mtb* upon exposure to AG. Transcriptomic analyses provide genome-wide insights into the variations in gene expression in a given sample with respect to its reference condition. Whereas it gives a general idea of the number of differentially expressed genes (DEGs) and their broad functional categories, it does not discriminate between genes linked to the condition directly or indirectly and those that are not. Thus it is not helpful in obtaining mechanistic insights into condition-specific variations. Stochasticity is a commonly observed feature in gene expression variations and hence typically there are a large number of genes that do not correlate directly with the specific condition of a given sample. A further complexity arises as even among those DEGs correlated to the given condition, it is difficult to distinguish between genes that cooperate with each other to result in a coherent response versus those that are variations explored by the cell to overcome any stress that the cells may be exposed to in the given condition. Further, while deciding which gene is differentially expressed, for practical reasons, a rather arbitrary cut-off, such as a fold change of 1.5–2.0 folds, is chosen and all those below the cut-off that may still be significant are lost. These limitations can be overcome by using a molecular networks approach that harness the power of computational approaches to combine condition-specific expression data with general PPI map to construct dynamic and stress response networks.

We have previously established a methodology for mapping transcriptome data onto the PPI network and mining the network to identify sets of nodes that form connected sets, or in other words, sub-networks that are most varied in the a given condition with respect to its reference (*Sambarey et al., 2013*; *Sambaturu et al., 2016*). This is analogous to identifying where the major differences are in a city's traffic pattern at two different time points. Only those DEGs that show significant alteration in their neighborhoods are picked in the top-ranked sub-networks and hence the method is very useful in identifying the few hubs from which many other variations could emanate in the network. It must be noted that DEGs are not always directly connected with each other. Instead, there are a number of bridging nodes, some of which are constitutively expressed, and connect DEGs, leading to trails or paths that mediate flow of some sort of signal. Altogether, combining global expression profile of *Mtb* in response to AG with the PPI revealed both putative signal(s) and pathways activated in response to AG exposure.

