## [Decision Letter]

Thank you for submitting your article "Efficacy of β-lactam/β-lactamase inhibitor combination is linked to WhiB4 mediated changes in redox physiology of *Mtb*" for consideration by *eLife*. Your article has been favorably evaluated by Philip Cole (Senior Editor) and three reviewers, one of whom, Bavesh D Kana (Reviewer #1), is a member of our Board of Reviewing Editors. The following individual involved in review of your submission has agreed to reveal their identity: Carl Nathan (Reviewer #2).

The reviewers have discussed the reviews with one another and the Reviewing Editor has drafted this decision to help you prepare a revised submission.

Summary:

In this study, the authors attempt to describe the role of the redox sensor, WhiB4, in mediating the transcriptional and metabolic response of *Mycobacterium tuberculosis* to combinatorial treatment with Amoxicillin and a β-lactamase inhibitor, clavulanate. These agents were administered in the clinically familiar form of Augmentin (AG). The authors start their exposition by describing the transcriptional response of *M. tuberculosis* to AG, combined with some in silico network analysis, and describe several transcriptional changes that suggest perturbations in peptidoglycan (PG) biogenesis, β-lactamase regulation, membrane permeability and efflux pumps. Further interrogation of these responses suggested changes in carbon metabolism and respiration, including differential expression of genes in the TCA cycle and various components of the electron transport chain. The authors also identify a set of responses that point to changes in redox physiology. They further explore these redox-associated changes by interrogating gene expression in AG-exposed mutant strains of *M. tuberculosis* defective in Msh (mycothiol) and Egt (ergothioniene), key regulators in redox balance. Next, they demonstrate that AG exposure induces redox stress in the form of dysregulated NADH/NAD ratios and the accumulation of reactive oxygen intermediates. Quenching of reactive oxygen species rescues AG-mediated killing. Developing this further, the authors use an MSH-specific redox reporter and demonstrate that oxidative stress accumulated in *M. tuberculosis* when treated with AG in axenic culture and in macrophages. To substantiate these effects, the authors next turn to study the effects of AG treatment in a strain defective for MSH and show that this leads to a differential tolerance of AG-mediated killing effects. Thereafter, the authors study the transcriptional response to AG in a WhiB4 defective strain and report substantively increased expression of genes involved in β-lactam tolerance and PG biosynthesis, the latter confirmed by changes in the spatial localization of PG biosynthesis in the WhiB4 mutant. Using different approaches, the authors then demonstrate that WhiB4 is able to regulate β-lactamase expression in a redox-dependent manner and that loss thereof, affects the inherent susceptibility of *M. tuberculosis* to AG, a notion they further explore and confirm in mice. The authors conclude that WhiB4 may act as a novel regulator of β-lactamase expression, an effect which is sensed through perturbations in oxidative-phosphorylation when *M. tuberculosis* is challenged with AG.

The manuscript reports a substantive body of carefully conducted experiments and opens up new avenues of investigation regards β-lactam treatment of TB disease. However, it in its current form, there are several shortfalls that need to be addressed.

Essential revisions:

1) A substantive amount of interpretative power is placed on transcriptional analysis when bacteria are exposed to 10X MIC of AG. Under these conditions, it is not surprising that with nascent damage to peptidoglycan, the organism's ability to maintain a proton gradient, ATP generation and a reductive intracellular state will be highly compromised. As such, the high concentration of drug used can obfuscate the underlying biology. To validate their observations, the authors are requested to repeat their transcriptional analysis at 1X and 5X the MIC of AG. Under these conditions, transcriptional changes reflecting those seen at 10X MIC would validate their observations.

2) As bacteria are being treated with a cell wall inhibitor, growth and death should be monitored by CFU. Changes in cell shape can be misleading when monitoring by OD. The authors are requested to conduct CFU analysis in all cases where killing/survival was measured by OD.

3) EMSA assays need a cold-competitive DNA control.

4) In the Materials and methods section, the sentence, "Various concentrations of AG were added immediately after infection and at various time points macrophages were…." is somewhat concerning. If this is the way the experiment was done, how did the authors confirm that infection was indeed established?

5) The manuscript requires substantive editorial overhauling. As it currently stands, it is very dense, does not flow well and is hence not suitable for the broad readership base of *eLife*. For some guidance, the network analysis and transcriptional data reported in the main body are complicated and difficult to follow. The authors are requested to lift out the key aspects of the network analysis and present them in the main body. The rest must be moved as high quality, explanatory figures to the supplementary information. Figure 1 can remain as is. Please endeavor to ensure that information in the supplementary information is of equivalent quality and flow as that placed in the main text. Also, a carefully drawn, visually appealing model that describes the authors findings would be useful. Perhaps the mouse data can be presented as a time course on one graph? This is standard in the field.

---

## [Author Response]

Essential revisions:

1) A substantive amount of interpretative power is placed on transcriptional analysis when bacteria are exposed to 10X MIC of AG. Under these conditions, it is not surprising that with nascent damage to peptidoglycan, the organism's ability to maintain a proton gradient, ATP generation and a reductive intracellular state will be highly compromised. As such, the high concentration of drug used can obfuscate the underlying biology. To validate their observations, the authors are requested to repeat their transcriptional analysis at 1X and 5X the MIC of AG. Under these conditions, transcriptional changes reflecting those seen at 10X MIC would validate their observations.

We thank the reviewers for these insightful suggestions. Although our results show that exposure to 10X MIC of AG does not exert lethality at 6 h post-treatment [Figure 1—figure supplement 1-Inset; revised manuscript], we completely agree that high concentrations of AG can adversely affect *Mtb*’s physiology to complicate interpretation of drug-specific effects on gene expression. Therefore, it is very important to validate these observations at lower concentrations of AG. To specifically address this concern, we repeated microarray experiments by exposing *Mycobacterium tuberculosis (Mtb*) to 1X and 5X MIC of AG for 6 h and 12 h, post-treatment (GEO accession number GSE93091). We first confirmed that exposure to 1X and 5X MIC of AG does not compromise the viability of *Mtb* at 6 h and 12 h post-treatment by enumerating colony-forming units (CFUs) [Figure 1—figure supplement 1-Inset; revised manuscript]. Further, microarray data revealed that a relatively small number of genes were differentially regulated at 1X and 5X MIC as compared to 10X MIC of AG [[Supplementary-material SD5-data]; revised manuscript]. However, similar to our earlier observations with 10X MIC, exposure to 1X and 5X MIC of AG increased expression of genes associated with PG biogenesis, β-lactamase regulation, cell envelope stress, antioxidant systems, alternate respiration, central carbon metabolism (CCM), and efflux pumps [Figure 2—figure supplement 4; revised manuscript].

Therefore, expression changes at various concentrations of AG are in reasonable agreement with each other. Finally, microarray data was validated by performing qRT-PCR on a few genes highly deregulated upon treatment with 1X, 5X, and 10X MIC of AG [Figure 2—figure supplement 1, revised manuscript]. These results are incorporated in the revised manuscript (subsection “AG affects pathways associated with central carbon metabolism (CCM), respiration, and redox balance”, last paragraph).

2) As bacteria are being treated with a cell wall inhibitor, growth and death should be monitored by CFU. Changes in cell shape can be misleading when monitoring by OD. The authors are requested to conduct CFU analysis in all cases where killing/survival was measured by OD.

3) EMSA assays need a cold-competitive DNA control.

This is an excellent suggestion and we appreciate this recommendation. Accordingly, we performed competition assays using *blaC* and *blaR* upstream sequences as positive controls, while promoter fragment of an unrelated gene *Rv0986* was utilized as a negative control.

First, we performed competition assays using 10, 25, 50, 100, and 200-fold molar excess of *blaC* and *blaR* DNA fragments. We found that 100-fold molar excess of *blaC* and *blaR* fragments completely prevented WhiB4 binding. However, the same concentration of an unlabeled *Rv0986* promoter fragment was inefficient to out-compete WhiB4 association with *blaC* and *blaR* DNA fragments [Figure 8—figure supplement 1; revised manuscript]. These results are included in the revised manuscript and we thank the reviewers for these helpful suggestions.

4) In the Materials and methods section, the sentence, "Various concentrations of AG were added immediately after infection and at various time points macrophages were…." is somewhat concerning. If this is the way the experiment was done, how did the authors confirm that infection was indeed established?

We apologize for not adequately explaining the technical details of this experiment. These are now included in the *Materials and methods* section of the manuscript. To determine the effect of AG during infection, PMA-differentiated THP-1 cells were infected with wt *Mtb* at a multiplicity of infection of 10 (MOI:10) to triplicate wells. Infection was allowed to establish by incubating macrophages for 4 h at 37°C and 5% CO_2_. At 4 h post-infection, macrophages were washed and resuspended in RPMI medium containing 200 µg/ml amikacin for 2 h to remove extracellular bacteria. Macrophages were then washed to remove amikacin and infection was confirmed by lysing macrophages using 0.06% SDS and enumerating released bacteria by serial dilution and plating on 7H11-OADC plates. At this point, infected macrophages were resuspended in fresh RPMI medium containing various concentrations of AG. At various time points post-exposure to AG, macrophages were lysed and released bacteria were serially diluted and plated on 7H11-OADC agar medium for CFU determination. We thank the reviewer for this comment.

5) The manuscript requires substantive editorial overhauling. As it currently stands, it is very dense, does not flow well and is hence not suitable for the broad readership base of eLife. For some guidance, the network analysis and transcriptional data reported in the main body are complicated and difficult to follow. The authors are requested to lift out the key aspects of the network analysis and present them in the main body. The rest must be moved as high quality, explanatory figures to the supplementary information. Figure 1 can remain as is. Please endeavor to ensure that information in the supplementary information is of equivalent quality and flow as that placed in the main text. Also, a carefully drawn, visually appealing model that describes the authors findings would be useful. Perhaps the mouse data can be presented as a time course on one graph? This is standard in the field.

We greatly appreciate these recommendations and believe that it has considerably improved our manuscript. Based on the reviewer’s suggestions, we have made significant changes in the manuscript to simplify complex data and to clearly explain key findings in the main manuscript. The changes are as outlined below:

i) Network analysis is clearly described in the revised manuscript. Detailed explanation and methodology used for network construction is provided in the Materials and Methods section of the revised manuscript].

ii) The sub-networks, which are related to the central theme of the manuscript such as cell wall processes and intermediary metabolism/respiration, are described in the main body of the manuscript. Relevant high quality images of sub-networks along with explanatory legends are provided in the supplementary information [Figure 1—figure supplement 2; revised manuscript].

iii) We also agree with the suggestion to model key findings of our manuscript as a visually appealing model. Therefore, in the revised manuscript we have summarized our findings to provide a model showing redox basis of AG tolerance in *Mtb* [Figure 10; revised manuscript].

iv) As suggested, mouse data is represented as a time course[Figure 9; revised manuscript]. For better clarity two graphs have been made. Figure 9 shows CFU data of wt *Mtb, MtbΔwhiB4, whiB4-comp*, and *whiB4-OE* strains, whereas Figure 9 displays data obtained for the XDR strains *MYC 431* and *431/whiB4-OE*.

v) We have made efforts to clearly write our findings with an easy to follow rationale along with experimental details. Specifically, we removed unnecessary and complicated information to revamp some portions of Introduction, Results, and Discussion. Particularly, results pertaining to NADH/NAD^+^ measurements, *E_MSH_* measurements, growth phenotypes of various redox-altered mutants, and β-lactamase regulation by WhiB4 are now explained in a lucid and concise manner. Importantly, data generated from the new experiments are summarized in simple terms in a few sentences in the revised manuscript.